# Improved Order Tracking in Vibration Data Utilizing Variable Frequency Drive Signature

**DOI:** 10.3390/s25030815

**Published:** 2025-01-29

**Authors:** Nader Sawalhi

**Affiliations:** Defence Science and Technology Group (DSTG), Melbourne, VIC 3207, Australia; nader.sawalhi@defence.gov.au

**Keywords:** variable frequency drive (VFD), pseudo tachometer, order tracking, bearing faults

## Abstract

Variable frequency drives (VFDs) are widely used in industry as an efficient means to control the rotational speed of AC motors by varying the supply frequency to the motor. VFD signatures can be detected in vibration signals in the form of sidebands (modulations) induced on tonal components (carrier frequencies). These sidebands are spaced at twice the “pseudo line” VFD frequency, as the magnetic forces in the motor have two peaks per current cycle. VFD-related signatures are generally less susceptible to interference from other mechanical sources, making them particularly useful for deriving speed variation information and obtaining a “pseudo” tachometer from the motor’s synchronous speed. This tachometer can then be employed to accurately estimate the speed profile and to facilitate order tracking in mechanical systems for vibration analysis purposes. This paper presents a signal processing technique designed to extract a pseudo tachometer from the VFD signature found in a vibration signal. The algorithm was tested on publicly available vibration data from a test rig featuring a two-stage gearbox with seeded bearing faults operating under variable-speed conditions with no load, i.e., with minimal slip between the induction motor’s synchronous and actual speed. The results clearly demonstrate the feasibility of using VFD signatures both to extract an accurate speed profile (root mean square error, RMSE of less than 2.5%) and to effectively perform order tracking, leading to the identification of bearing faults. This approach offers an accurate and reliable tool for the analysis of vibration in mechanical systems driven by AC motors with VFDs. However, it is important to note that some inaccuracies may occur at higher motor slip levels under heavy or variable loads due to the mismatch between the synchronous and actual speeds. Slip-induced variations can further distort tracked order frequencies, compromising the accuracy of vibration analysis for gear mesh and bearing defects. These issues will need to be addressed in future research.

## 1. Introduction and Background

### 1.1. Order Tracking

Frequency analysis of vibration signals is commonly used to evaluate the condition of rotating machinery. For this analysis to yield meaningful results, each vibration component must correspond to a distinct frequency. In rotating machinery, most vibration components are dependent on the speed of the drive shaft. When the shaft speed is constant, discrete frequency values are observed. However, when shaft speed fluctuates, frequency analysis becomes ineffective, as discrete frequency values no longer exist. To overcome this, the vibration signal can be re-sampled from the time domain to the phase domain. Since the shaft rotates the same angular distance in each revolution, re-sampling the signal to phase ensures that vibration components always correspond to discrete frequency values, regardless of speed variations. This transformation produces an order-spectrum, where frequency analysis remains valid even for machines with variable speeds. The technique used to generate this order-spectrum is known as order tracking [1]. Order tracking has proven successful in detecting and diagnosing various mechanical faults, including gear faults, misalignment and bearing defects, and is commonly utilized for gear fault detection [2], misalignment fault diagnosis [3], and bearing fault diagnosis [4]. For effective order tracking, a speed reference signal synchronized with the vibration measurements is essential. Typically, these signals are acquired using speed sensors such as tachometers, key phasors (proximity probes), or optical encoders. Speed sensors may be impractical or challenging to install in harsh operating conditions [5,6], or they may be limited to a location that may not be synchronously related to a shaft of interest. In such situations, “tacho-less” approaches have been developed that rely solely on vibration signals to deduce the instantaneous angular speed. Various tacho-less order tracking techniques have emerged to convert vibration signals into the order domain from the time domain.

Narrow-band phase demodulation techniques, such as those suggested by Bonnardot et al. [7], employ the acceleration signal at specific gear mesh frequencies to track small variations in rotational speed. Another common technique works on extracting a “pseudo” tachometer signal through the phase demodulation of a shaft harmonic component after bandpass filtering, providing a phase (shaft rotation angle) versus time map that is sampled at uniform intervals [7]. With larger speed variations, it often becomes necessary to demodulate a low-frequency harmonic to avoid contamination from sidebands related to another harmonic [7]. Another, relatively straightforward and simplistic, method to extract a pseudo-tacho involves identifying a separable band in the frequency domain that contains one of the harmonics of the shaft of interest, extracting it (as a complex frequency signal with phase intact) into a new buffer, and inverse transforming it back into the time domain [8,9]. Zero crossing detection can then be used to identify shaft rotations and obtain a speed profile. The vibration signal can then be resampled (order tracked) with the aid of these zero crossing locations to ensure a constant number of samples for each shaft rotation. This method is illustrated schematically in Figure 1.

The “pseudo-tacho” algorithm relies on discrete Fast Fourier Transform (FFT) and inverse FFT operations, making it practical and easily adaptable for industrial applications. The DFT transforms a time-domain signal *x*[*n*] into its frequency-domain representation *X*[*k*] using Equation (1):(1)$$X\left[k\right]=\sum_{n=0}^{N-1}x\left[n\right].e^{-j\frac{2\pi }{N}kn}\;,\;k\;=\;0,\;1,\ldots ,N-1$$
where:*X[k]:* Frequency-domain representation of the signal[*n*]: Time-domain signal*N*: Total number of samples (FFT size)*k*: Frequency index (0 to *N−*1)*n*: Time sample index (0 to *N*−1)*j*: Imaginary unit (sqrt(−1))

The inverse FFT (IFFT) reconstructs the time-domain signal from its frequency components as provided by Equation (2):(2)$$X\left[n\right]=\frac{1}{N}\sum_{k=0}^{N-1}x\left[k\right].e^{j\frac{2\pi }{N}kn}\;,\;n\;=\;0,\;1,\ldots ,N-1$$

These computations are efficiently performed using software tools such as MATLAB, R2023a which provide built-in functions for FFT and IFFT, ensuring accurate and fast implementation.

Spectrogram-based methods have gained popularity, with Heyns et al. [10] introducing a clustering procedure to track instantaneous frequency (IF), filtering it to resample the signal for fault detection. In cases of larger speed variations, maxima-tracking approaches have demonstrated efficacy. Urbanek et al. [11], Zhao et al. [6] and Schmidt et al. [12] developed techniques to extract the instantaneous frequency from spectrograms that employ filtering and resampling to track signals under varying speed conditions. Others have employed advanced signal decomposition and transformation techniques; e.g., He et al. [13] used discrete spectrum correction, and Leclere et al. [14] used a multi-order method utilizing probability density functions. Significant advancements in ridge detection and time-frequency analysis have also been achieved. Li et al. [15,16] introduced the improved cost function ridge detection (ICFRD) method, which enhances ridge extraction in time-frequency planes through adaptive bandwidth and cost function optimization. Legros et al. [17] combined short-time Fourier transforms and sparse sampling theory to estimate signal modes in noisy environments. Li et al. also proposed the automated and adaptive ridge extraction (AARE) method [18], and the time-frequency ridge estimation (TFRE) method [19].

In addition to “tacho-less” order tracking methods, various techniques exist for diagnosing and analyzing bearing faults without speed data, which is particularly important at higher slip levels. Entropy-based approaches, such as those presented by Leite et al. [20], offer effective solutions for early fault detection. Recent advancements in bearing fault detection also include return map analysis, which utilizes time-series feature extraction enhanced with wavelet transform techniques, as demonstrated by Ponomareva et al. [21]. These methods enable accurate detection and analysis of bearing faults even in the absence of direct speed data.

### 1.2. Variable Frequency Drives (VFDs)

Despite advancements in tacho-less order tracking methods, accurately identifying and isolating a separable frequency band based on one of the rotor shaft harmonics remains a challenging task due to interference from various other frequency components that are not related to the speed. To address this issue in machinery with VFDs, it is proposed to exploit the VFD signature embedded in the vibration signal as a means to extract an accurate estimate of the AC supply frequency, i.e., the motor’s instantaneous synchronous speed. For induction motors, the shaft speed will approximate the synchronous speed under low-load conditions (minimum slippage).

VFDs play a vital role in applications requiring precise rotational speed and torque control of an AC electric motor. A typical VFD includes an AC rectifier, a DC link, and a DC-to-AC inverter. The rectifier initially converts the AC input voltage—characterized by constant amplitude and frequency—into a pulsating DC voltage, often using a full-wave bridge rectifier with diodes [22]. Following this, the DC link regulates and smooths the rectified signal, typically employing a capacitor bank for energy storage. This arrangement stabilizes the DC voltage and mitigates load-side transients that could affect the distribution system. The DC link serves as a bridge between the rectifier and the inverter, enabling the conversion of the smoothed DC back into a variable-frequency AC supply, often employing advanced technologies such as insulated-gate bipolar transistors (IGBTs). The inverter utilizes Pulse Width Modulation (PWM) techniques to produce a pseudo-sinusoidal voltage of varying frequency [23]. Typically, a high-frequency triangular or sawtooth carrier wave is compared with a sinusoidal signal with the desired output amplitude and frequency. The IGBT switches on when the signal exceeds the carrier wave and switches off when the signal falls below it. The resulting PWM output is then typically passed through a d*V*/d*t* and/or a sinewave filter to provide a smoother AC output with the desired voltage and frequency.

VFDs can operate at various carrier frequencies, such as 4, 6, 8, and 10 kHz. Users operating test rigs at lower (e.g., 4 or 6 kHz) frequencies often experience high-pitched noise [24]. Conversely, increasing the carrier frequency to 8 or 10 kHz significantly reduces this audible noise. Generally, higher carrier frequencies lead to a more sinusoidal waveform, albeit at the expense of increased losses within the VFD. The vibration characteristics resulting from VFD operation typically include harmonics of the carrier frequency with sidebands at both the rotor speed and VFD frequency. Notably, the magnitude of the carrier frequency and its harmonics is generally lower than that of the sidebands. Higher-amplitude sidebands manifest at the pseudo line frequency of the VFD, with their amplitude increasing as the carrier frequency decreases [24].

### 1.3. Envelope Analysis

A comprehensive tutorial on rolling element bearing diagnostic analysis can be found in [25] and is highly recommended for providing a clear overview of how bearing faults manifest in vibration signals and how diagnostic information can be extracted. In bearing diagnostics, a primary challenge is that the raw signal spectrum often provides limited insight into potential faults, prompting the need for alternative methods to detect faults in the early stages. Over time, envelope analysis—also known as the high-frequency resonance technique (HFRT)—has become the standard approach for diagnosing bearing issues. The process involves three key steps:Bandpass Filtering (Capturing a band where impulses are amplified by structural resonance): The raw signal is filtered within a high-frequency band to isolate fault-induced impulses. Although these impulses are weak in the original signal, they are amplified by structural resonances at higher frequencies.Amplitude Demodulation (Highlighting amplitude indicative of fault characteristics): The filtered signal is then amplitude-demodulated to generate the envelope signal. This step extracts variations in amplitude that are indicative of fault characteristics.Spectrum Analysis of the Envelope Signal: The spectrum of the envelope signal contains critical diagnostic information, including:Repetition frequencies (such as ball pass frequency or ball spin frequency), which correspond to the rotation of bearing elements.Modulation frequencies, which reflect the rate at which the fault interacts with the load zone or moves relative to the measurement point.

Although envelope analysis was initially developed using analog methods, significant improvements have been made through the use of digital processing techniques. A major advancement has been the application of Hilbert transform techniques for amplitude demodulation. This method involves:Taking a one-sided spectrum (positive frequencies only),Applying an inverse transformation to the time domain, resulting in a complex time signal known as the “analytic signal”. The imaginary part of this signal is the Hilbert transform of the real part.

A key benefit of the Hilbert transform approach is its ability to more precisely select and demodulate the desired frequency band using an ideal filter. This enables better separation of fault-related frequencies from stronger components, such as gear mesh frequencies. In contrast, analog filters and real-time digital filters often face limitations in their ability to effectively filter out unwanted signals [26].

Envelope analysis works well in cases of incipient faults and rolling contact fatigue (RCF), which produce impulses in the system. It requires the identification of a suitable band for demodulation. However, the absence of RCF limits its effectiveness, as bearing defect frequencies are not expected to appear in the envelope spectrum. In such cases, statistical health indicators relying on signal impulsiveness, such as Kurtosis, are not suitable. Instead, monitoring friction level increases, noise, and energy changes in the system would be more appropriate.

### 1.4. Paper Contribution and Structure

The primary contribution of this paper is demonstrating the feasibility of using VFD sidebands for order tracking and fault diagnosis, rather than providing an exhaustive comparison to other technologies. This paper is organized as follows: Section 2 (Materials and Methods) presents a summary of the public dataset and the experimental test rig (two-stage gearbox driven by a VFD three-phase four-pole motor). Two separate approaches for extracting speed profiles are then discussed. The first examines the VFD signature from the current signal, while the second utilizes the VFD signature from the vibration signal. Section 3 (Results) details the use of VFD signatures in vibration data for identifying inner and outer race faults in the system. Section 4 (Summary and Discussion) summarizes the findings of the paper and sheds light on key insights and findings presented in the paper. Finally, Section 5 (Conclusions and Future Work) provides paper conclusions and proposes some future work investigations.

## 2. Materials and Methods

### 2.1. Experimental Setup and Data Collection

This study utilizes a public dataset from a vibration test rig powered by a small 3 HP (2.24 kW), 3-phase, 4-pole induction motor controlled by a VFD [27]. For a 4-pole motor, the synchronous speed, $\omega_{sync-speed}$, can be calculated using Equation (3):(3)$$\omega_{sync-speed}\left(Hz\right)=\frac{2\times VFD}{n_{poles}}=\frac{2\times VFD}{4}$$

The details of the test rig are illustrated in the schematic presentation of Figure 2. Reference [27] has the full details and depicts an image of the test rig. The motor connects to a two-stage step-up gearbox with a reported overall gear ratio of 2.07 (see gear teeth discussion below). The dataset comprises various data types, including vibration data, motor current data, acoustic data, and temperature data, all collected under different load conditions, speeds, and fault scenarios. For variable-speed conditions, only vibration, motor current, and speed data at no load are available. Speed is measured at the gearbox output high-speed shaft (HSS) and ranges from 680 to 2460 RPM. The vibration data are sampled at 25.6 kHz, the current data at 100 kHz, and the acoustic data at 51.2 kHz. The speed data was collected using a tachometer (Autonics FD-620-10) positioned on the HSS shaft. Instead of raw once-per-revolution data, RPM values for the HSS were provided at 0.11 s intervals. Additionally, constant-speed data are available at a motor speed of approximately 1500 RPM, resulting in a gearbox output speed of about 3010 RPM.

A schematic representation of the two-stage gearbox is shown in Figure 3. The exact number of teeth on each gear was not provided. These were therefore deduced using the vertical vibration signal recorded at bearing A under constant-speed conditions (HSS ≈ 3010 RPM). This corresponds to a motor and gearbox low-speed shaft (LSS) speed of 1498 RPM based on the reported overall gear ratio (2.07) of the gearbox. The method of identifying the number of teeth was based on the approach outlined by Sawalhi and Randall [28] in their study on gear parameter identification using vibration signals. The identified number of teeth for each gear were calculated to be, N1 = 60, N2 = 40, N3 = 58, and N4 = 42, giving an overall actual gear ratio of 87/42 ≈ 2.0714. The corresponding actual frequencies for the LSS, intermediate-speed shaft (ISS), and HSS are therefore 24.22 Hz (1453.22 RPM), 36.33 Hz (2179.80 RPM), and 50.17 Hz (3010.2 RPM), respectively. Figure 4 shows the vibration spectrum of bearing A (vertical direction) under constant speed conditions. The second-stage gear mesh frequency (1 × GMF2) is clearly identified at 2106.43 Hz, corresponding to N4 × HSS. The second harmonic of the first gear mesh frequency (2 × GMF1) appears at 2905.43 Hz, with an amplitude higher than the fundamental (1 × GMF1), suggesting a possible misalignment of the LSS. Further evidence of LSS misalignment is provided by the distinct sidebands around 1 × GMF1 (1452.715 Hz) at 1 × LSS, as displayed in the zoomed-in section. Additionally, the vibration signal includes the variable frequency drive (VFD) carrier frequency at 6002.4 Hz, with noticeable sidebands at 2 × VFD. These VFD-related observations are discussed further in Section 2.2 and Section 2.3.

In the varying load condition test, each bearing fault—inner race, outer race, and ball fault—was introduced separately in bearing housing B. The bearing used for testing is an NSK 6205 DDU, featuring a ball diameter (*d*) of 7.90 mm, a pitch diameter (*D*) of 38.5 mm, a contact angle (*θ*) of zero degrees, and a total of 9 balls (*n*).

Equations (4)–(7) were used to calculate bearing defect frequencies for various components as follows:

Ball Pass Frequency Inner Race (BPFI): Frequency at which balls pass a point on the inner race.(4)$$BPFI=\frac{nf_{r}}{2}\cdot \left(1+\left(\frac{d}{D}\right)\cdot cos\left(\theta \right)\right)$$

Ball Pass Frequency Outer Race (BPFO): Frequency at which balls pass a point on the outer race.(5)$$BPFO=\frac{nf_{r}}{2}\cdot \left(1-\left(\frac{d}{D}\right)\cdot cos\left(\theta \right)\right)$$

Ball Spin Frequency (BSF): Rotational frequency of the rolling elements themselves.(6)$$BSF=\frac{d}{2D}\cdot \left(1-{\left(\frac{d}{D}\cdot cos\left(\theta \right)\right)}^{2}\right)\cdot f_{r}$$

Fundamental Train Frequency (FTF): Frequency at which the cage rotates relative to the bearing.(7)$$FTF=\frac{1}{2}\cdot \left(1-\left(\frac{d}{D}\right)\cdot cos\left(\theta \right)\right)\cdot f_{r}$$
where:*n*: Number of rolling elements*d*: Diameter of rolling elements*D*: Pitch diameter of the bearingθ: Contact angle (in radians)$f_{r}$: shaft rotational speed (Hz, RPM or 1 for orders)

Table 1 presents the fundamental train frequency (FTF), ball pass frequency inner (BPFI), ball pass frequency outer (BPFO), and ball spin frequency (BSF), expressed as orders of the high-speed shaft (HSS).

### 2.2. Current and Vibration Signal Comparison Under Constant Speed

Figure 5 presents a comparison between the current and vibration spectra from bearing housing A in the low-frequency range (0–150 Hz), displayed on both linear and logarithmic (dB) scales. From Figure 5 we can see that the VFD is 48.4812 Hz (motor synchronous speed = VFD/2 = 24.2406 Hz) and that the LSS = 24.2133 Hz. As such, the slip frequency can be calculated as 24.2406−24.2133 = 0.0273 Hz (1.638 RPM). In the current spectrum, the dominant frequency is 1 × VFD at 48.4812 Hz, which is clearly visible in the linear scale. In the log-scale (dB) spectrum, two frequencies at 24.2756 Hz and 72.6876 Hz are also visible in addition to 2 × VFD. These two frequencies are very close to 0.5 × VFD and 1.5 × VFD. However, they appear to be spaced at approximately 24.2056 Hz around the VFD, suggesting a modulation of the VFD by the rotor speed, as the spacing is closer to the rotor speed (24.2133 Hz) rather than 0.5 × VFD (24.2406 Hz). The vibration spectrum provides more detailed information about the mechanical system. The 1 × HSS frequency at 50.1567 Hz is clearly visible, along with a significant 2 × HSS harmonic, which appears to suggest some misalignment of the HSS. Both the ISS (36.32 Hz) and LSS (24.2133 Hz) frequencies are also present in the linear spectrum, but with lower amplitudes compared to the HSS. In the log-scale spectrum, components at 2×, 3×, and 4 × LSS can also be seen.

Figure 6 compares the current and vibration linear spectra around the VFD carrier frequency. In the current spectrum, sidebands are visible at 1 × VFD, with the VFD carrier itself exhibiting a low amplitude. In contrast, the vibration spectrum shows the VFD carrier along with prominent sidebands at 2 × VFD. This behavior is attributed to the motor’s magnetic forces (torque) fluctuating at twice the VFD frequency due to two magnetic flux peaks per current cycle. Note that a smaller amplitude vibration at 1 × VFD sidebands is also evident. This is possibly due to slight variations between the pole pairs in the 4-pole motor. Torque modulation typically takes place around the steady-state torque component (0 Hz) and the VFD PWM carrier frequency. Modulation around the steady-state torque frequency may overlap with other mechanical vibrations, such as harmonics from the motor or gearbox. However, modulation near the VFD carrier frequency is more isolated, allowing for clearer demodulation of the VFD frequency.

### 2.3. Speed Profile Extraction from the Current Signal Under Variable Speed Operation

Figure 7 illustrates a typical current signal and its frequency content for variable-speed operation under no-fault conditions. In the signal shown in Figure 7a, a varying sinusoidal frequency is observed, with a fundamental frequency corresponding to the VFD frequency. This can be further examined using frequency analysis (Figure 7c), which shows that the VFD frequency varies between 10 and 40 Hz. For a 4-pole motor, the motor synchronous speed (≈LSS under no load) will therefore vary between 5 to 20 Hz. The HSS speed is the LSS (motor) speed multiplied by the gear ratio of 87/42. Zooming in on the time interval between 0.1 and 0.11 s (Figure 7b) highlights the sawtooth carrier frequency at 6 kHz, accompanied by voltage switching spikes. Zooming in on the frequency interval around the 6 kHz carrier frequency (Figure 7d) shows that the first-order VFD sidebands around the 6 kHz carrier in fact carry the same information as the VFD baseband (Figure 7c) and can therefore relay the same information.

A pseudo tachometer signal can be derived from any of the three frequency bands illustrated in Figure 7: Band A (0–50 Hz), Band B (5950–6002.4 Hz), or Band C (6002.4–6050 Hz). It is important to note that the carrier frequency at 6002.4 Hz represents a line of symmetry, as indicated by the red dashed line in Figure 7d. This line is analogous to zero frequency. Consequently, in Figure 7d, the frequencies to the right of this line (Band C) represent positive frequencies, while those to the left (Band B) represent negative frequencies. To extract a speed profile, a buffer filled with zeros, sized similarly to the Fast Fourier Transform (FFT) size, is first created. The frequency band of interest (one of the three bands mentioned above) is then copied into this buffer, ensuring that the carrier frequency (for Bands B and C) aligns with the zero frequency. Following this, the buffer undergoes an inverse FFT transformation to the time domain, resulting in a pseudo encoder signal at the VFD line frequency. By identifying the zero crossings of this pseudo encoder signal, the instantaneous VFD frequency can be calculated. To obtain a speed profile for the HSS, the VFD frequency is multiplied by 2 and divided by 4 as per Equation (3). This yields the synchronous motor speed (≈LSS), which is then multiplied by the gear ratio (87/42) to obtain the HSS speed. This process is illustrated schematically in Figure 8 in the cases of using either current or vibration data. Note that in the case of using vibration data, the bands B and C represent a sideband at 2 × VFD, and thus 2 × VFD is directly calculated by dividing the sampling frequency by the sample difference between the zero crossings.

The extracted speed profiles for the three bands are displayed in Figure 9, with smoothing applied using a median filter of 5 samples. Importantly, the use of any of these bands yields a matching speed profile as compared to the actual RPM profile, as there is no significant signal interference present. The estimated RPM (synchronous speed) was compared to the actual rotor RPM by interpolating both to a common time axis. Cross-correlation was applied to align the RPM values, ensuring synchronization between the actual and estimated data. Relative error percentages were calculated using Equation (8) and plotted in Figure 10. It is important to note that the comparison involves the estimated RPM based on synchronous motor speed, which naturally differs from the actual RPM due to motor slippage. The highest overall root mean square error (RMSE), 4.64%, was observed in Band A, more than four times the error seen in Band B (1.01%) and Band C (1.05%). The maximum error across all bands occurs around the 40 s mark, where a sharp speed change coincides with the lowest RPM values. The error arises from a combination of slippage, no loading condition, and the smoothing effect caused by the median filter.(8)$$Relative\;Error\;\left(\%\right)=\frac{Estimated\;RPM-Actual\;RPM}{Actual\;RPM}\times 100\%$$

Time-frequency plots alongside the extracted speed profiles (with no smoothing) are presented in Figure 11 and Figure 12 for Bands A, B and C, respectively. Note that these two figures show time-frequency plots for the selected bands, clearly illustrating the observed speed variations. Furthermore, the extracted speed profiles align closely with the variations depicted in these time-frequency plots.

### 2.4. Speed Profile Extraction Using VFD Signature in the Vibration Signal

Figure 13 presents a 60 s acceleration signal measured on Bearing B (vertical direction) accompanied by a spectrogram that reveals the speed profile variation. This variation is clearly observable in the low-frequency regions, around the gear mesh frequencies, and near the 6 kHz VFD carrier frequency. Figure 14 further examines the vibration signal collected from Bearing B within three frequency bands [0–45 Hz], [5900–6002.4 Hz], and [6002.4–6100 Hz], designated as Bands A, B and C respectively. It is important to note that to extract a tachometer reference, one must isolate a frequency band that contains the variation from a single component without interference from other components. This task is particularly challenging in this context, as different shaft harmonics often coexist in the selected frequency band. The situation is further complicated in the vicinity of the gear mesh frequencies due to the rich content of sidebands surrounding each harmonic. Note that Bands B and C are asymmetric (unlike the equivalent signal bands in the current signal presented earlier in Figure 7d). Asymmetric sidebands in the vibration signal from a variable frequency drive (VFD) carrier frequency can result from several factors. The use of Pulse Width Modulation (PWM) leads to non-linearity in the switching pattern, causing uneven energy distribution around the carrier frequency. Note that the sidebands in the current signal are more balanced because the inductive nature of the load (such as motors) naturally filters out high-frequency PWM harmonics, smoothing the current waveform. In contrast, the vibration signal is more directly affected by the switching and mechanical resonances, leading to more pronounced and uneven sidebands around the carrier frequency. Variations in load, such as changes in torque or inertia, can further contribute to a non-linear frequency response and asymmetry.

The extracted speed profiles for the three bands are presented in Figure 15 with smoothing applied using a median filter of 21 samples. Notably, the speed profile derived from Band A fails to provide an accurate representation of the actual speed, particularly at lower speeds. In contrast, the speed profiles obtained from Bands B and C clearly yield accurate results that closely match the measured RPM profile. Figure 16 presents a quantitative comparison of the estimated RPM values to the actual RPM in terms of percentage errors. The results indicate that Band A has the highest error, with an RMSE of 58.82%, demonstrating its poor performance. Band B significantly reduces the error, achieving an RMSE of just 1.42% and a maximum absolute error of 7.88%, highlighting its improved reliability. Band C, although slightly higher than Band B with an RMSE of 2.13%, still offers a highly accurate representation of the actual RPM overall, despite showing poorer performance in the low-speed vicinity around the second 40.

Figure 17 and Figure 18 further examine the RPM estimated results by displaying spectrogram plots for the selected bands, alongside the actual and smoothed profiles for Bands A, B, and C, respectively. Note that Band A is dominated by the 1 × HSS and 2 × HSS harmonics, not the VFD, and so the speed profile in this particular case is based on the demodulation of these components. Figure 17 illustrates the interaction between the 1 × HSS and 2 × HSS harmonics under misalignment conditions, highlighting the greatest error occurring at minimum speed and the underlying cause of the poor performance in estimating the actual RPM and the high percentage of errors seen. In Figure 18, the spectrogram around the VFD carrier frequency reveals a clear speed profile at 2 × VFD, characterized by minimal interference, thereby resulting in a more accurate estimate of the speed profile, which was observed by the low RMS percentage of error.

## 3. Results

### 3.1. Inner Race Fault Identification

Figure 19 shows the time and frequency plots of a vibration signal with an inner race fault. The frequency bands used for HSS speed profile extraction are presented in Figure 19c (Band A) and Figure 19d (Bands B and C: sidebands at 2 × VFD). Note that Bands B and C are asymmetric, with more energy levels seen in Band C.

Figure 20 shows the speed profile extracted using Band A [0–45 Hz]. The unfiltered RPM profile exhibits spikes where the HSS fundamental interacts with the 2 × HSS harmonic. Applying a median smoothing filter with a window of 21 samples effectively eliminates these spikes. The smoothed RPM profile can be seen to closely follow the pattern observed in the spectrogram, except for the peaks, which are flattened due to the smoothing filter. Figure 21 displays the RPM speed profiles (both raw and smoothed), extracted using the VFD signature in Bands B and C, alongside their corresponding spectrograms. The profile obtained from Band C, which has higher energy, is notably less susceptible to interference, providing a clear representation that closely aligns with the RPM pattern observed in the spectrogram. In contrast, the RPM profiles derived from Band B are less accurate, particularly at peak speeds, where the lowest negative left sideband frequencies exhibit the least energy and, as such, are more affected by noise interference.

Overall, the results indicate that using VFD sidebands, especially those with higher energy such as Band C, yields better outcomes in terms of error levels at low RPM with sudden speed changes. This is further confirmed by examining the percentage of errors presented in Figure 22. This figure shows the error percentage in estimating HSS RPM for Bands A, B, and C. Band A exhibits the highest RMSE (3.59%), with notable spikes in error between 35–40 s during periods of low RPM, likely due to interference within the band. Bands B and C have lower RMSEs, 1.41% and 1.26%, respectively, indicating better accuracy and robustness.

Figure 23 presents a comparison of the residual envelope signals and spectra for order-tracked signals using Bands A, B, and C. The results indicate that order tracking with VFD Bands B and C significantly outperforms that using Band A. This is evident in the number of identified harmonics, the reduction of the smearing effect, and the clarity of the sidebands around the ball pass frequency of the inner race (BPFI). Band C delivers the clearest envelope spectrum, effectively illustrating the BPFI without any smearing up to the fourth harmonic. Additionally, the sidebands around the BPFI, particularly the 2× harmonic indicative of misalignment, are prominently visible. Band B also performs well, though not as effectively as Band C, while Band A lacks clarity and harmonic identification. Overall, the analysis demonstrates that using VFD bands, especially Band C, enhances the detection and characterization of critical frequency components in the vibration signals.

### 3.2. Outer Race Fault Identification

Figure 24 presents the time and frequency plots of the vibration signal measured from a defective bearing with an outer race fault. The RPM speed profile extraction bands are illustrated in Figure 24c (Band A) and Figure 24d (Bands B and C). Notably, Bands B and C exhibit symmetry with similar energy levels, in contrast to the scenario with the inner race defect. This symmetry suggests that both bands are likely to yield effective RPM profile extraction with minimal noise interference. The disparity in spectral symmetry observed between the inner and outer race bearing faults may be explained by the different ways the vibration signals are influenced in each case. When there is spalling in the inner or outer race, impacts from the rolling elements at the spall location may excite different natural frequencies, which could alter the vibration spectrum. Specifically, for the inner race defect, the spall may move in and out of the load zone as the bearing rotates, causing modulations in the vibration signal. This interaction may generate more pronounced and uneven sidebands around the VFD carrier frequency. In contrast, for the outer race defect, the spall is typically stationary in the load zone, leading to a more continuous interaction with the rolling elements. This may result in more symmetrical sidebands and similar energy levels around the VFD carrier frequency.

Figure 25 shows the speed profile obtained using Band A [0–45 Hz]. The unfiltered RPM profile closely follows the spectrogram, with minimal interference from the 2 × HSS component, indicating that the speed variations are mainly influenced by the fundamental harmonic. When a median smoothing filter is applied, the smoothed RPM profile aligns well with the spectrogram pattern, though the peaks are flattened due to the filter effect. Overall, Band A appears to effectively capture the speed profile accurately.

Figure 26 shows the RPM speed profiles (both raw and smoothed) extracted from the VFD signature using Bands B and C, along with their corresponding spectrograms. Bands B and C exhibit similar characteristics due to similar energy levels and provide better speed profiles at the peaks, as these profiles are not flattened such as those from band A.

Figure 27 shows the error percentage for Bands A, B, and C. Band A has the highest RMSE of 3.84%, with significant spikes in error occurring at low RPM between 45 and 50 s. In contrast, Bands B and C achieve lower RMSEs, 0.92% and 0.98%, respectively, demonstrating greater accuracy and stability.

Figure 28 compares the residual envelope signals and spectra for order-tracked signals using bands A, B, and C. The results demonstrate that order tracking with VFD bands B and C significantly outperforms band A. This improvement is evident in the number of identified harmonics and the reduction of the smearing effect. Both bands B and C clearly display the fundamental ball pass frequency of the outer race (1 × BPFO) and its second harmonic, whereas band A only shows the 1 × BPFO at a low level and in a smeared manner.

## 4. Summary and Discussion

This study demonstrates the effectiveness of using VFD signatures in vibration signals to accurately extract speed profiles and perform order tracking under light load. A signal processing technique to extract a pseudo tachometer from VFD signatures in vibration signals has been presented. The algorithm has been tested on publicly available vibration data from a test rig with a two-stage gearbox, seeded bearing faults, and variable-speed conditions under no load, minimizing slip between synchronous and actual motor speeds. Results show that VFD signatures provide a reliable means of deriving speed variations directly from vibration data, especially in cases where traditional speed sensors are unavailable or insufficient due to rapid speed fluctuations. Speed profiles have been extracted from the vibration signal using a band around the fundamental frequency (Band A) and two additional bands around the carrier frequency: Band B (left) and Band C (right). The results for vibration signal Band A were dominated by the 1 × HSS and 2 × HSS harmonics rather than the VFD signature. Smoothing was applied using a median filter of 21 samples. It was observed that the speed profile derived from Band A did not accurately represent the actual speed, particularly at lower rotational speeds. In contrast, the speed profiles obtained from Bands B and C provided more accurate results, closely matching the measured RPM profile. These profiles also aligned well with those extracted from the current signal, where similar bands (A, B, and C) were used, further confirming their reliability. The use of VFD Bands B and C—representing the left and right sidebands at 2 × VFD around the carrier frequency—offers significant advantages for order tracking. In detecting both inner race and outer race faults, these bands not only present clearer representations of the ball pass frequencies (BPFI and BPFO) and their harmonics, but they also show fewer artifacts and improved harmonic identification compared to Band A, which relies on the HSS frequency. Bands B and C consistently yield the clearest envelope spectra, effectively capturing critical frequencies with minimal smearing. This enhanced performance is attributed to the higher energy content in the VFD sidebands of Bands B and C, making them more resilient to noise interference.

The VFD carrier sidebands method demonstrated its effectiveness as a low-cost, non-intrusive solution for speed estimation and vibration analysis under light load conditions. Its simplicity and lack of reliance on additional sensors make it a practical alternative to more complex methods. This approach offers accurate speed profiles and facilitates robust order tracking, making it a preferred choice for applications requiring reliable speed estimation around the VFD frequency. Direct comparison with other technologies was beyond the scope of this study, as the focus was to establish the feasibility and reliability of the proposed method.

It is important to note that as order tracking relies on a reference rotational speed, using the VFD as the reference may result in orders that do not align with actual rotor events, especially at higher slip levels under heavy or variable loads. Dynamic slip variations can further distort tracked order frequencies, reducing analysis accuracy. While synchronous and rotor speed profiles follow similar trends, the slip-induced offset prevents perfect alignment of rotor-related vibration signals, such as gear mesh and bearing defect frequencies, with the VFD-based reference. Therefore, accounting for slip-through adjustments or corrections is essential for accurate order tracking and reliable bearing frequency analysis in induction motors.

## 5. Conclusions and Future Work

This study primarily investigates the effectiveness of using a single frequency band around the VFD carrier frequency for two purposes:Extracting the rotor RPM profile under light or no-load conditions.Leveraging the extracted pseudo-tachometer information to perform order tracking, minimizing speed fluctuations, and generating an order-tracked squared envelope spectrum for diagnosing bearing faults.

The performance of the proposed method was benchmarked against the actual RPM profile. Spectrogram plots were used to clarify observations in the estimated RPM. To facilitate comparison, the estimated RPM was interpolated onto a common time axis with the actual RPM. Cross-correlation was employed to align the two profiles accurately. It is important to note that the comparison involves the estimated RPM based on synchronous motor speed, which naturally differs from the actual RPM due to motor slippage. A quantitative comparison of RMS percentage errors, summarized in Table 2, highlights the overall performance differences across three frequency bands: Band A, around the fundamental frequency, and two additional bands around the VFD carrier frequency—Band B (left) and Band C (right).

When using current signals, Band A demonstrated the poorest performance, with the highest RMSE of 4.64%. This error is more than four times that of Band B (1.01%) and Band C (1.05%). The largest discrepancies for Band A were observed between 40 and 45 s, corresponding to low RPM regions where interference likely caused inaccuracies. For vibration signals in the absence of faults, Band A again exhibited the highest RMSE (58.82%), indicating poor reliability. In contrast, Band B delivered the most accurate results with an RMSE of 1.42%, while Band C also performed well with an RMSE of 2.13%. Under inner race fault conditions, Band A produced the highest RMSE (3.59%), with prominent error spikes between 35 and 40 s during low RPM periods. Bands B and C yielded significantly lower errors of 1.41% and 1.26%, showcasing improved robustness and precision. For the outer race fault, Band A continued to underperform, recording the highest RMSE of 3.84% with notable spikes between 45 and 50 s at low RPM. Bands B and C achieved much lower RMSEs of 0.92% and 0.98%, respectively, demonstrating superior stability and accuracy.

In conclusion, Bands B and C, around the VFD carrier frequency, consistently outperformed Band A, in all tested conditions. These bands provided more reliable RPM profiles and robust order tracking capabilities, even under fault conditions, with RMSEs consistently below 2.5%.

Suggested future work should focus on testing and improving bearing fault detection methods under varying load conditions, particularly addressing slip-induced frequency shifts, such as those caused by speed-dependent loads (such as motor-driven fans). This should include evaluating the accuracy of order tracking when the VFD-extracted speed is used as the reference instead of rotor speed, assessing the reliability of envelope analysis for identifying bearing defect frequencies, and examining the impact of motor slip on fault detection performance. Other factors worth investigating include the interaction of drivetrain vibration components within the VFD carrier frequency bands, especially in cases where significant gear mesh frequencies overlap with the carrier region, as might occur in drivetrains with lower carrier frequencies (e.g., 4 kHz), and exploring demodulation around second or higher harmonics of the carrier frequency. These efforts aim to enhance the reliability and adaptability of vibration-based diagnostic techniques for diverse operational scenarios.

## Figures and Tables

**Figure 1 sensors-25-00815-f001:**
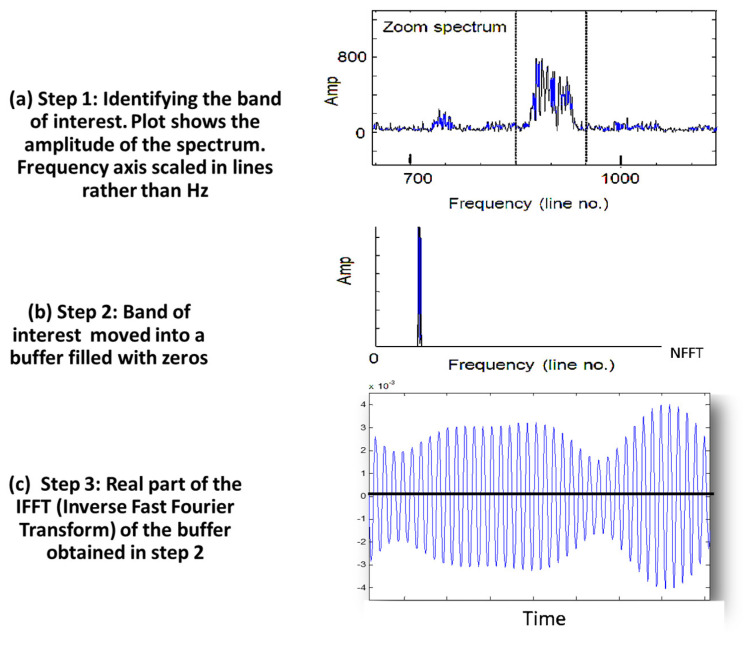
Reference (speed) signal extraction stages: (**a**) identifying a separable band, (**b**) extracting the band into a new buffer, and (**c**) inversing transforming signal b into the time domain (zoomed in) [9].

**Figure 2 sensors-25-00815-f002:**
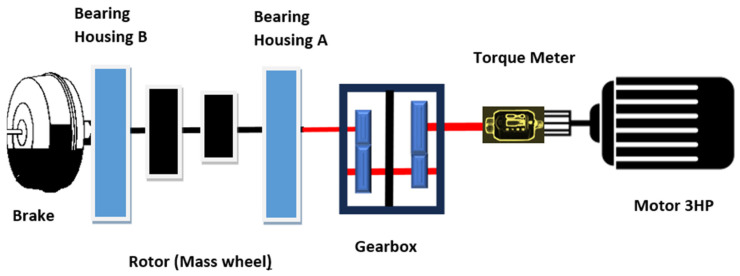
Schematic presentation of the test rig.

**Figure 3 sensors-25-00815-f003:**
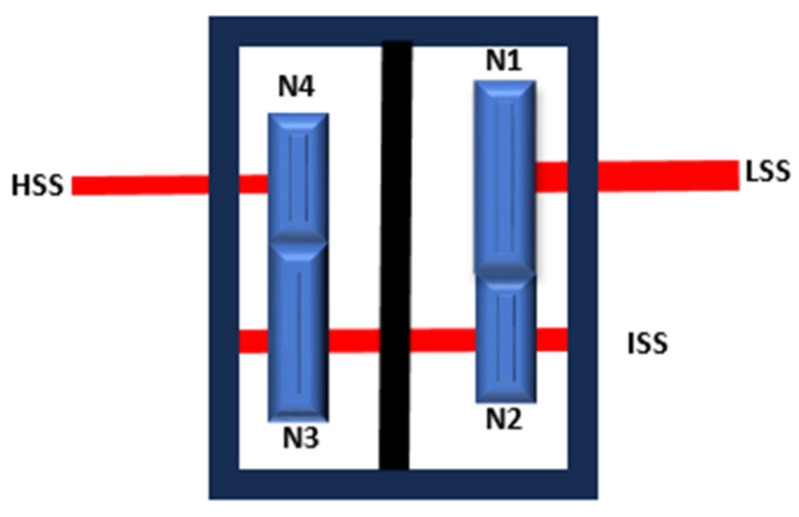
Schematic presentation of the 2-stage gearbox.

**Figure 4 sensors-25-00815-f004:**
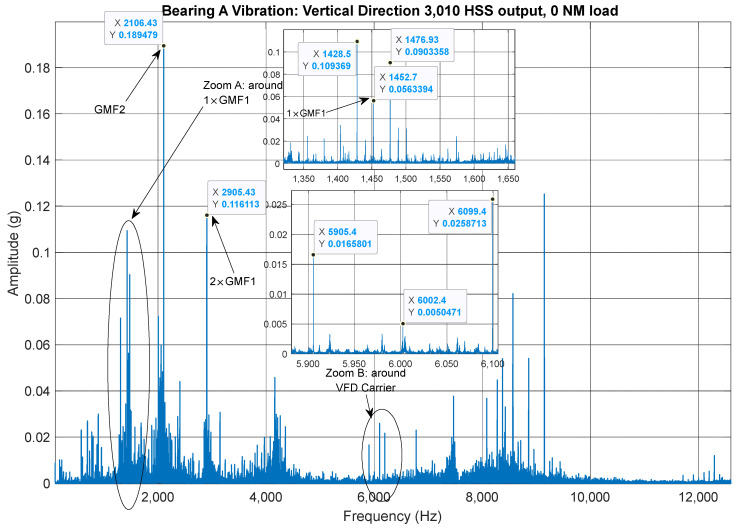
Bearing A vertical vibration under constant speed (HSS ≈ 3010 RPM) and no load. Zoom A around GMF2 [1350–1650] and Zoom B around 2 × GMF1 [5900–6100].

**Figure 5 sensors-25-00815-f005:**
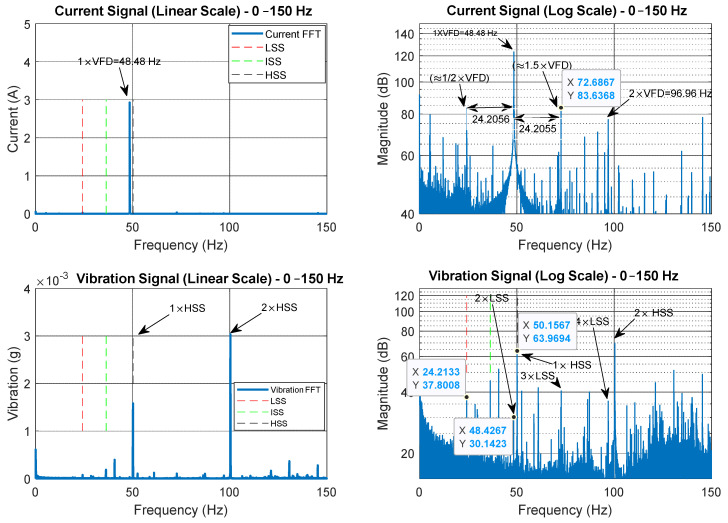
Comparison between current spectra and vibration spectra in the low-frequency region (0–150 Hz) on both linear and log (dB) scales.

**Figure 6 sensors-25-00815-f006:**
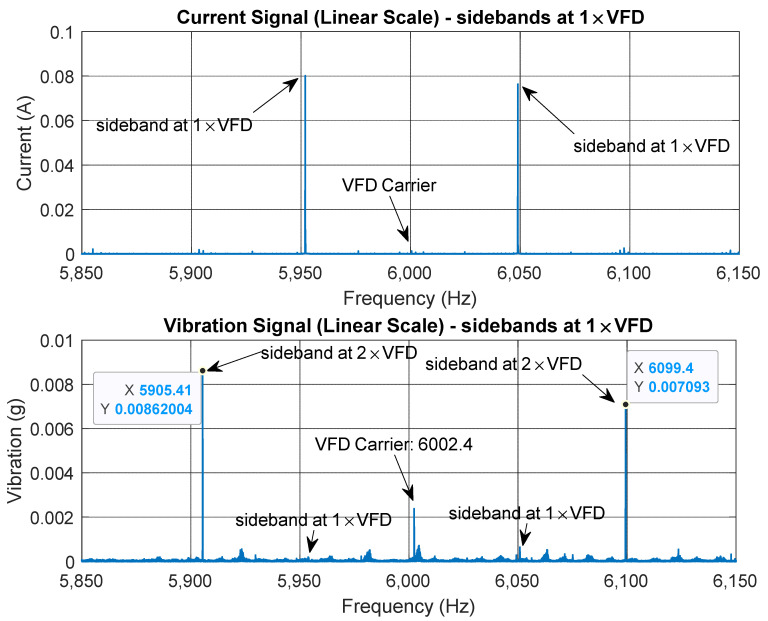
Comparison between current spectra and vibration linear spectra around the VFD carrier frequency.

**Figure 7 sensors-25-00815-f007:**
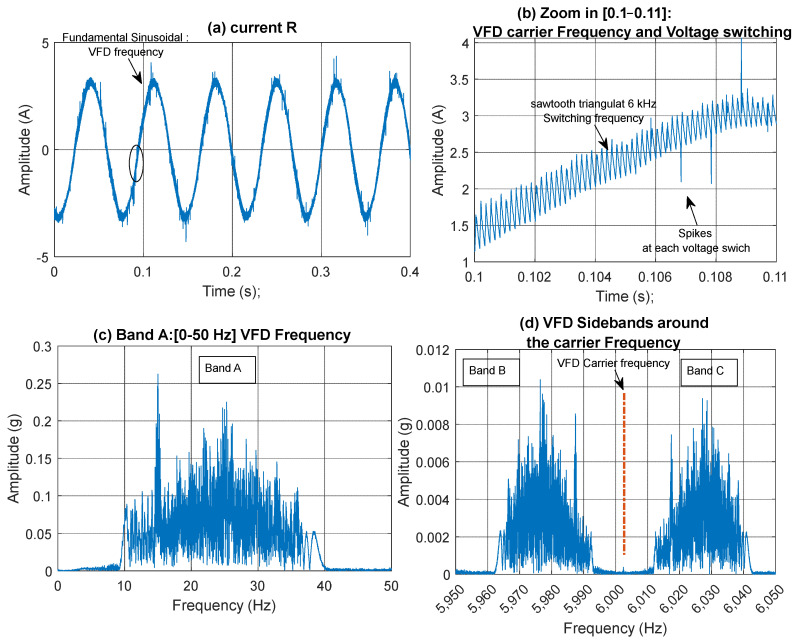
Current signal shown in the time and frequency domains: (**a**) time domain: circled 0.1–0.11 s; (**b**) time domain zoomed in [0.1–0.11] showing the VFD carrier frequency and voltage switching spikes; (**c**) frequency band A [0–50 Hz]; (**d**) first-order VFD sidebands around the carrier frequency: band B [5950–6002.4 Hz] and band C [6002.4–6050 Hz].

**Figure 8 sensors-25-00815-f008:**
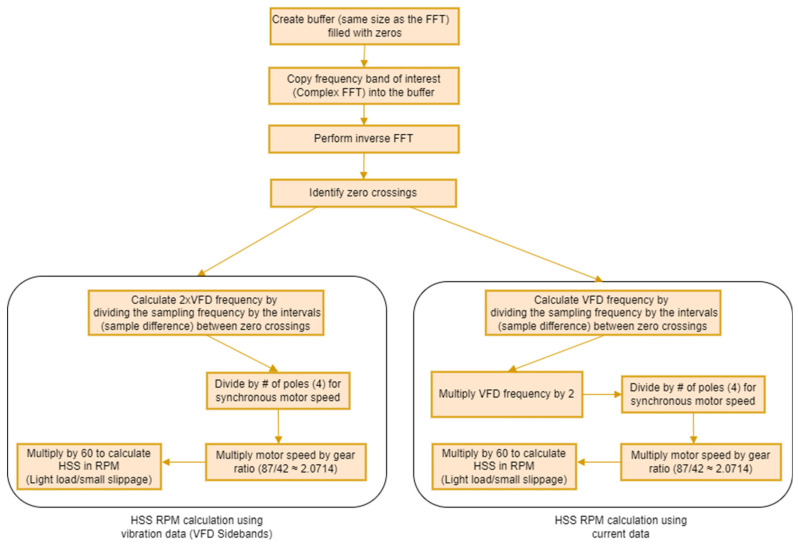
Pseudo tachometer extraction and RPM profile scaling using current and vibration data.

**Figure 9 sensors-25-00815-f009:**
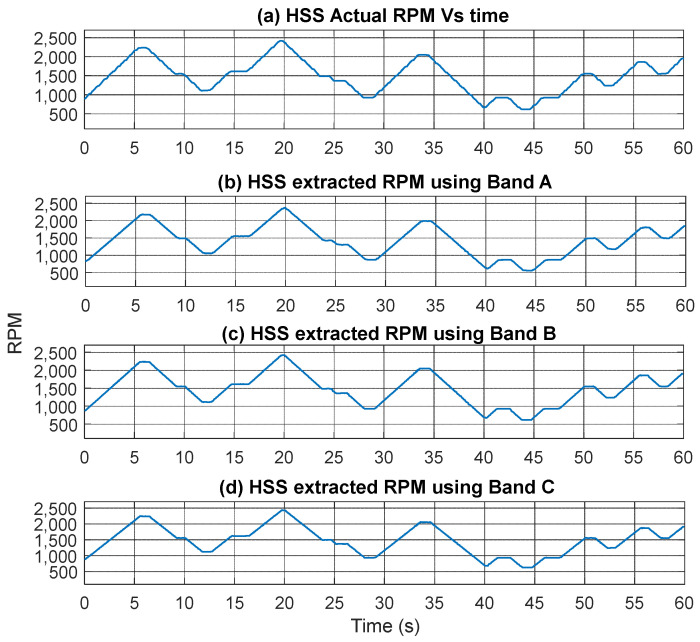
HSS speed-profile comparison: (**a**) actual RPM; (**b**) extracted RPM from VFD frequency: Band A; (**c**) extracted RPM from VFD carrier lower sideband: Band B; (**d**) extracted RPM from VFD carrier upper sideband: Band C.

**Figure 10 sensors-25-00815-f010:**
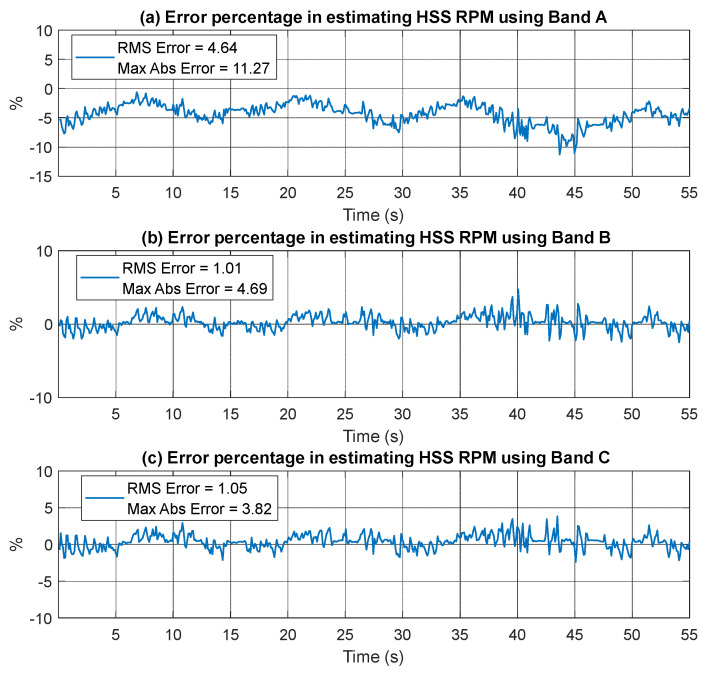
Error percentage in estimating RPM (fault-free case) using current signal (**a**) Band A (**b**) Band B and (**c**) Band C.

**Figure 11 sensors-25-00815-f011:**
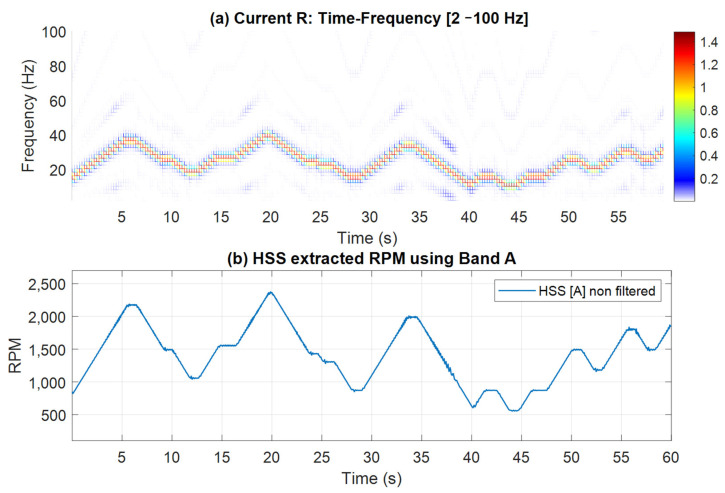
(**a**) Current time-frequency spectrogram [2–100 Hz]; (**b**) HSS extracted RPM profile using Band A without median filter smoothing.

**Figure 12 sensors-25-00815-f012:**
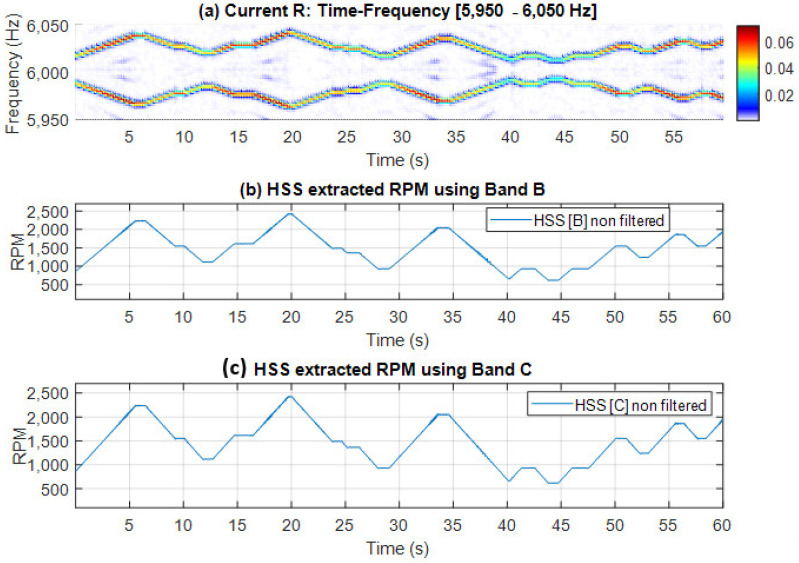
(**a**) Current time-frequency spectrogram [5950–6050 Hz]; (**b**) HSS extracted RPM profile using Band B without median filter smoothing; (**c**) HSS extracted RPM profile using Band C without median filter smoothing.

**Figure 13 sensors-25-00815-f013:**
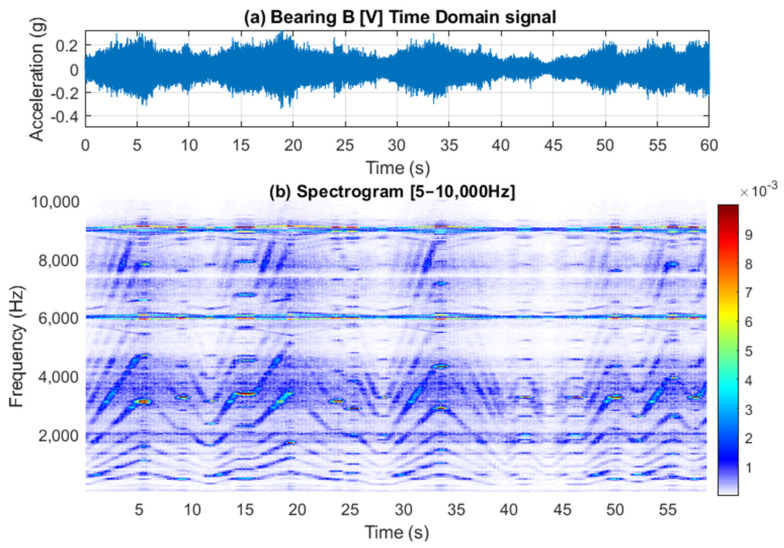
(**a**) Bearing B vertical vibration acceleration signal; (**b**) Spectrogram [5 Hz–10 kHz].

**Figure 14 sensors-25-00815-f014:**
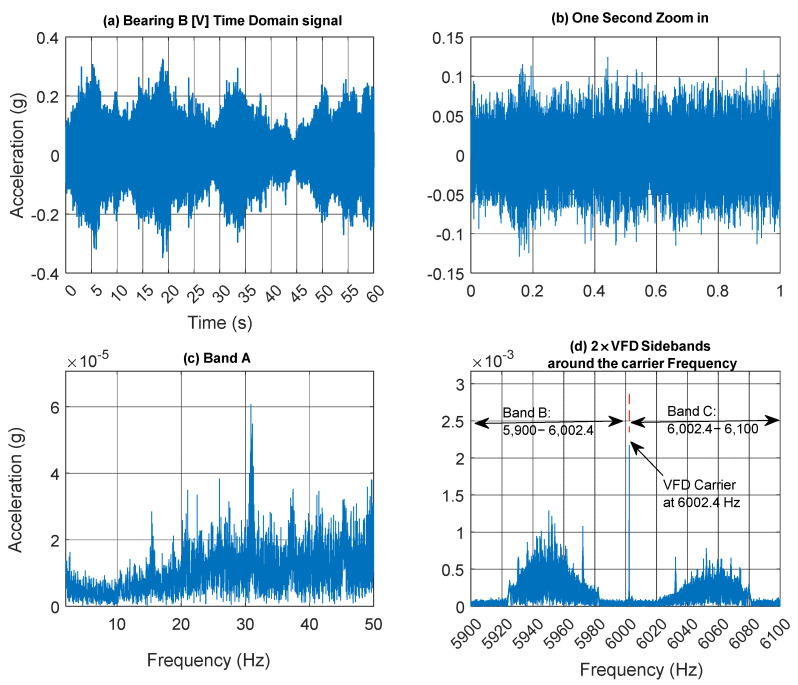
(**a**) Bearing B vertical vibration signal; (**b**) one second zoom in; (**c**) Band A; (**d**) Band B and C around the VFD carrier frequency.

**Figure 15 sensors-25-00815-f015:**
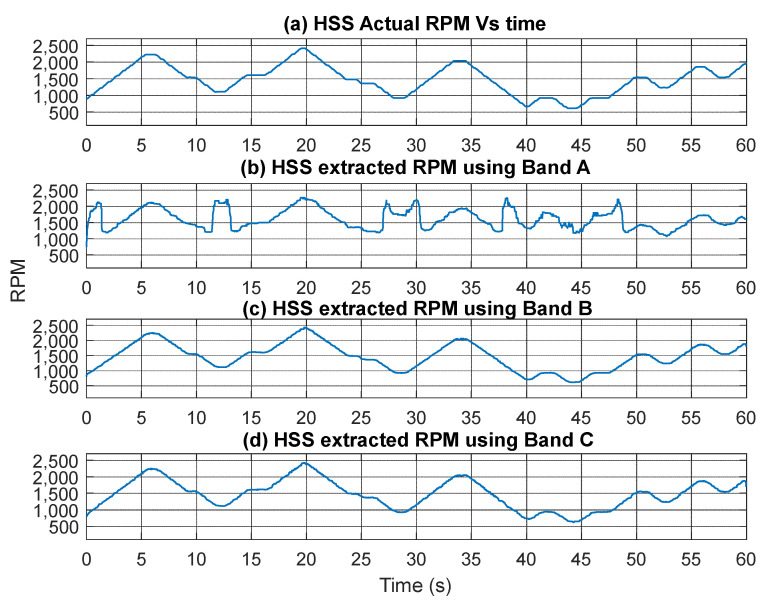
(**a**) HSS Actual RPM profile; (**b**) HSS extracted RPM profile using Band A; (**c**) HSS extracted RPM profile using Band B; (**d**) HSS extracted RPM profile using Band C.

**Figure 16 sensors-25-00815-f016:**
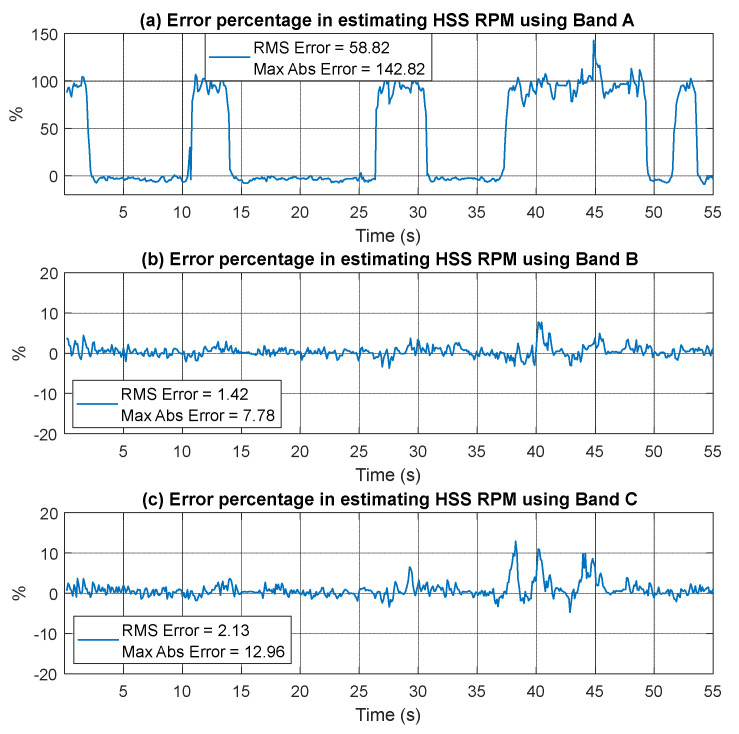
Error percentage in estimating RPM using vibration signa (fault-free bearings) based on (**a**) Band A, (**b**) Band B, and (**c**) Band C.

**Figure 17 sensors-25-00815-f017:**
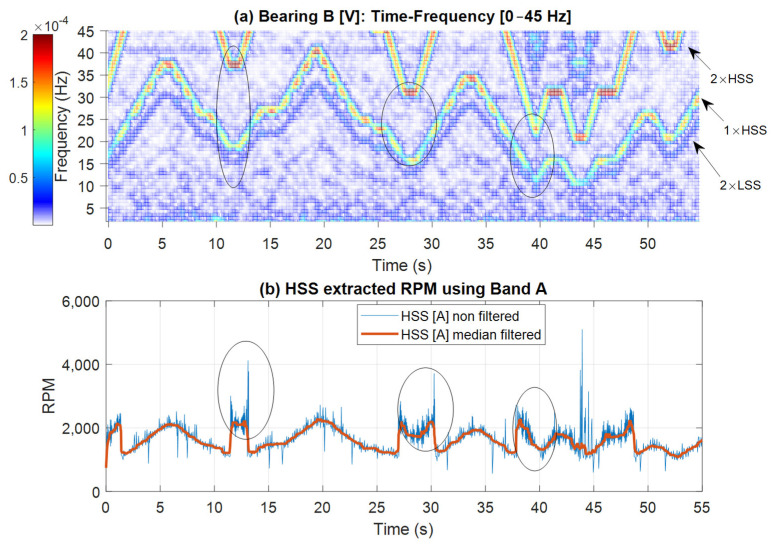
(**a**) Bearing B vertical vibration time-frequency plot; (**b**) HSS extracted RPM using Band A with and without smoothing. Circled: Low RPM regions with highest errors.

**Figure 18 sensors-25-00815-f018:**
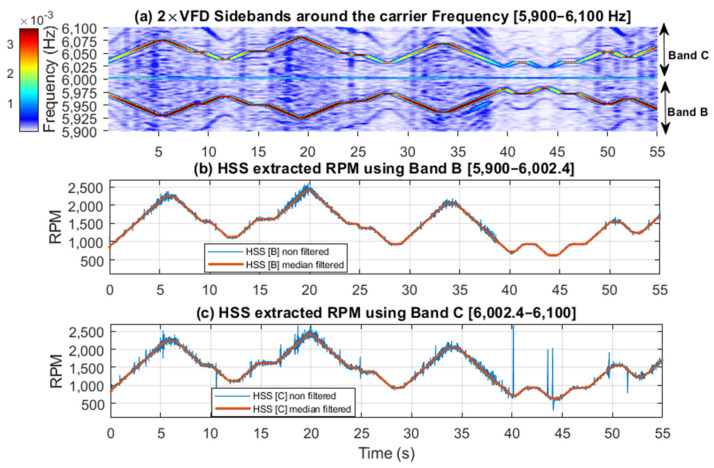
(**a**) 2 × VFD sidebands around the VFD carrier frequency showing Bands B and C; (**b**) HSS extracted RPM profile using Band B; (**c**) HSS extracted profile using Band C.

**Figure 19 sensors-25-00815-f019:**
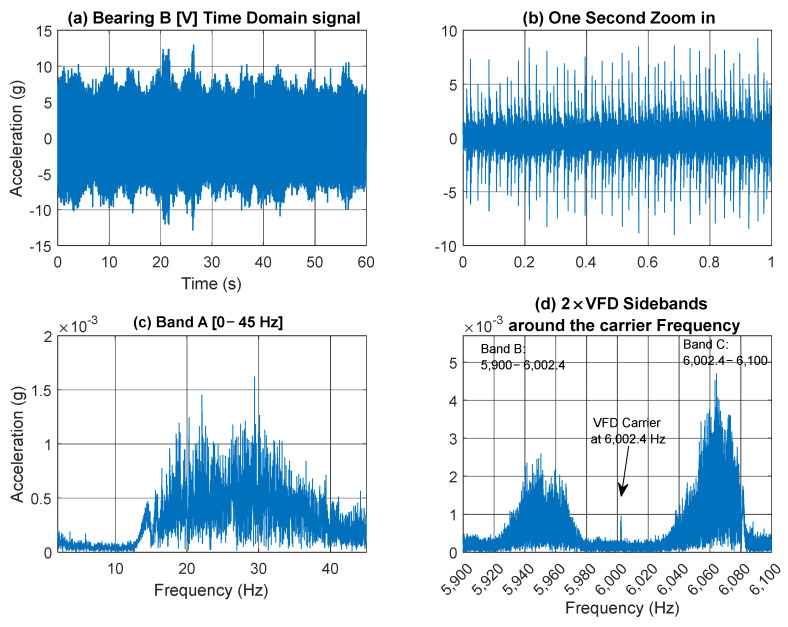
(**a**) Inner race HSS drive shaft vertical vibration signal; (**b**) one second zoom in; (**c**) frequency plot of the spectrum [Band A: 0–45 Hz); (**d**) zoom in around the carrier frequency [5900–6100 Hz) showing two bands: Band B the left of the carrier and Band C to the right.

**Figure 20 sensors-25-00815-f020:**
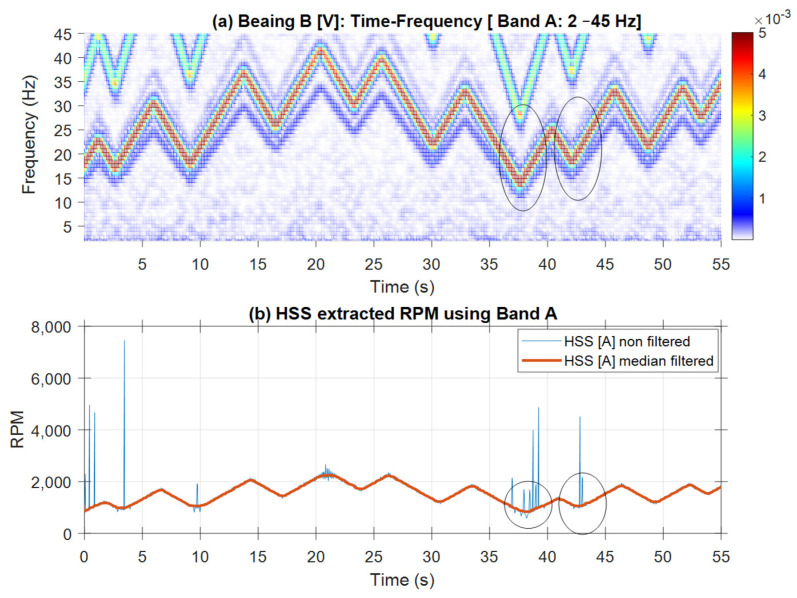
Inner race: (**a**) bearing B vertical vibration time-frequency plot (Band A: 2–45 Hz); (**b**) HSS extracted RPM using Band A with and without smoothing. Circled: low RPM with high interference and estimate error.

**Figure 21 sensors-25-00815-f021:**
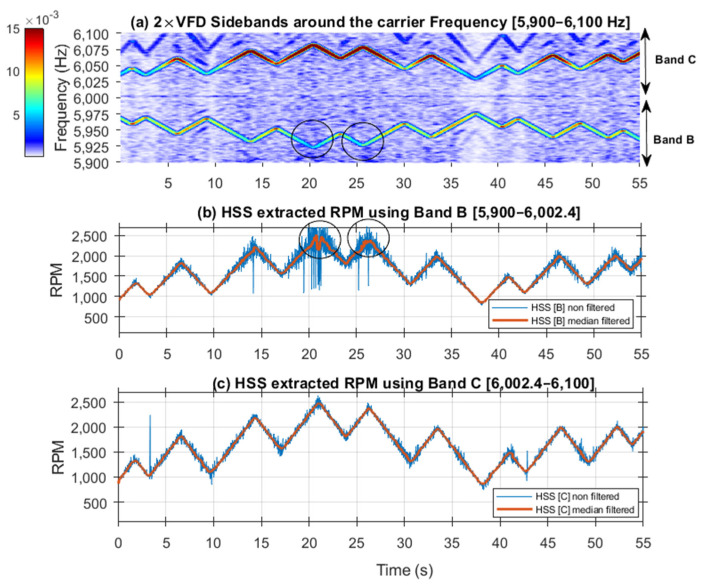
Inner race: (**a**) 2 × VFD sidebands around the VFD carrier frequency showing Bands B and C [5800–6200]; (**b**) HSS extracted RPM profile using Band B; (**c**) HSS extracted profile using Band C. Circled: Peak speeds, where Band B frequencies exhibit the least energy.

**Figure 22 sensors-25-00815-f022:**
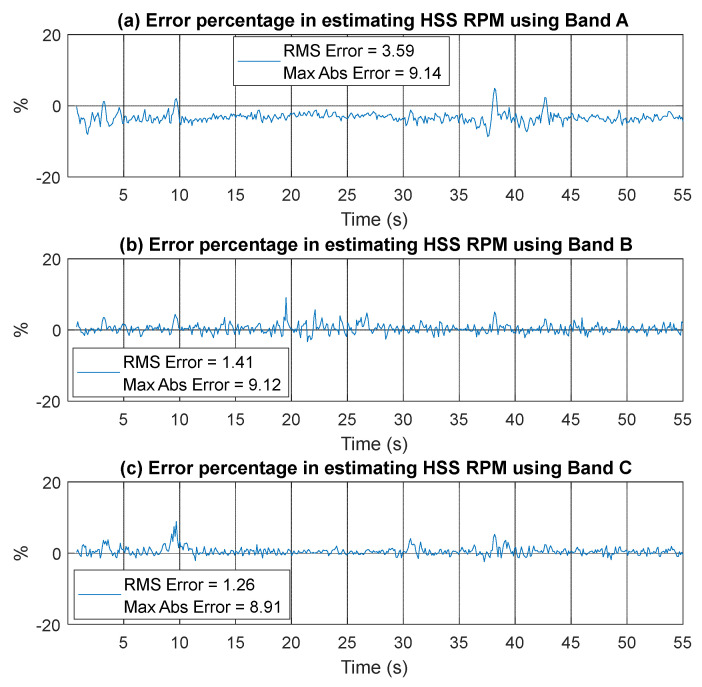
Error percentage in estimating RPM using vibration signal (inner race case) based on (**a**) Band A, (**b**) Band B, and (**c**) Band C.

**Figure 23 sensors-25-00815-f023:**
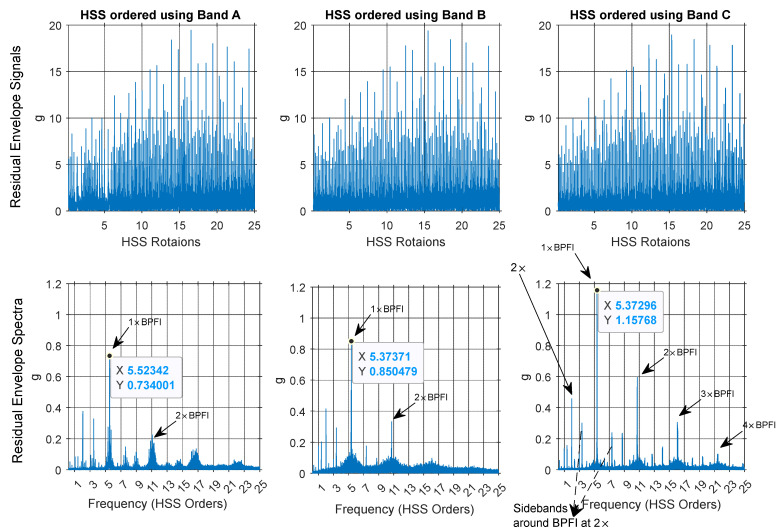
Inner race fault diagnosis: residual envelope signals and envelope spectra for Band A, Band B, and Band C.

**Figure 24 sensors-25-00815-f024:**
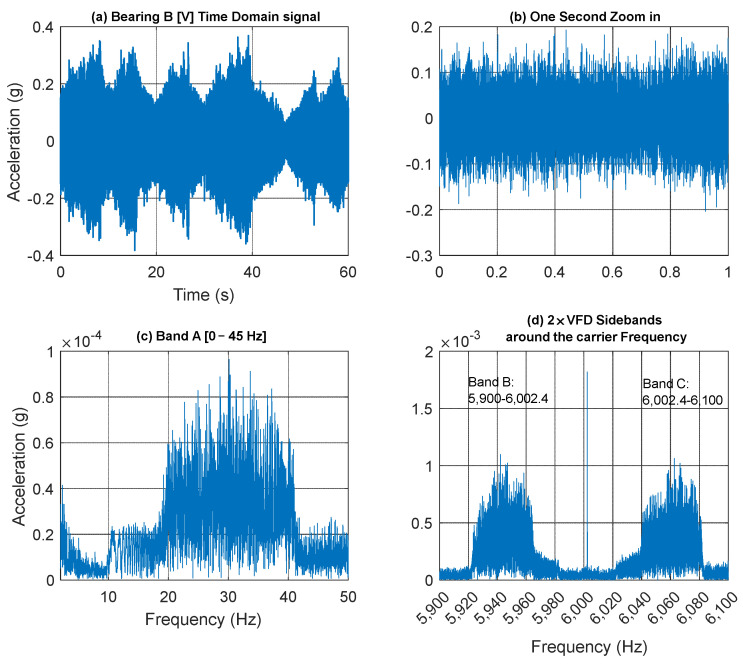
Outer race HSS drive shaft vertical vibration signal: (**a**) time domain; (**b**) one second zoom in; (**c**) zoom in plot of spectrum (Band A: 0–45 Hz); (**d**) zoom in around the carrier frequency (5900–6100 Hz) showing two bands: Band B to the left of the carrier and band C to the right.

**Figure 25 sensors-25-00815-f025:**
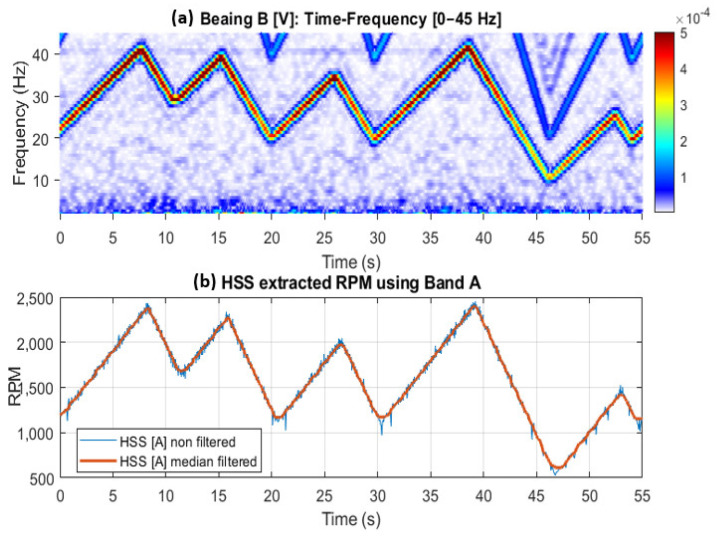
Outer race: (**a**) Bearing B vertical vibration time-frequency plot (Band A: 0–45 Hz); (**b**) HSS extracted RPM using Band A with and without smoothing.

**Figure 26 sensors-25-00815-f026:**
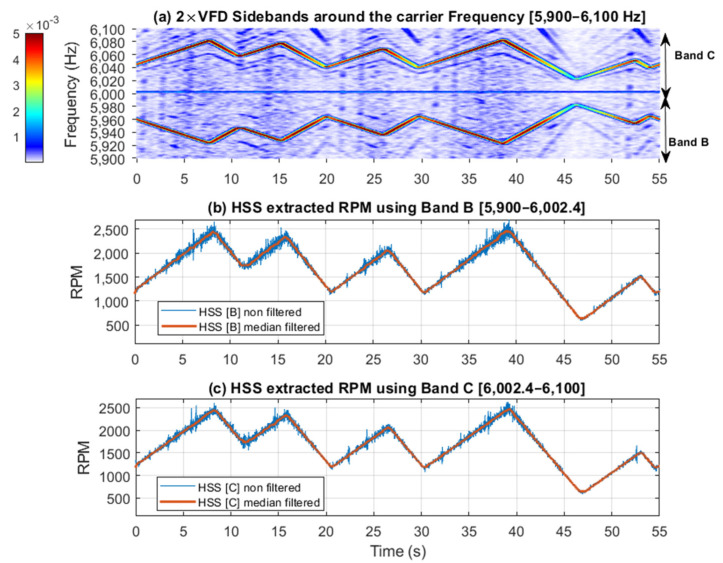
Outer race: (**a**) 2 × VFD sidebands around the VFD carrier frequency showing Bands B and C [5800–6200]; (**b**) HSS extracted RPM profile using Band B; (**c**) HSS extracted profile using Band C.

**Figure 27 sensors-25-00815-f027:**
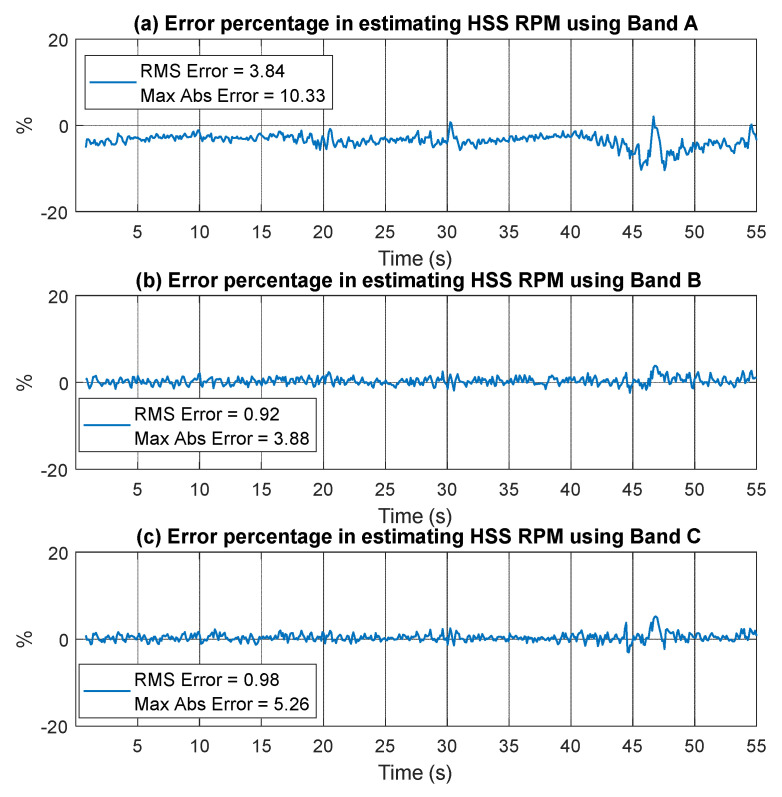
Error percentage in estimating RPM using vibration signal (outer race case) based on (**a**) Band A, (**b**) Band B, and (**c**) Band C.

**Figure 28 sensors-25-00815-f028:**
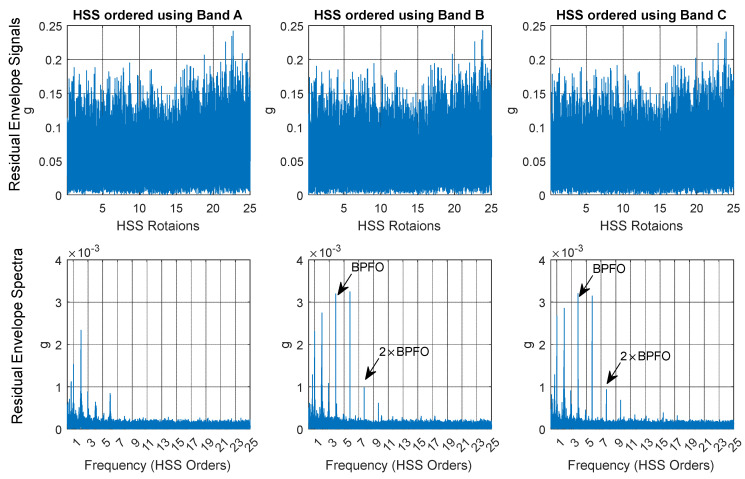
Outer race fault diagnosis: residual envelope signals and envelope spectra for Bands A, B, and C, respectively.

**Table 1 sensors-25-00815-t001:** Bearing fault frequencies.

Bearing Fault Frequencies	Orders (×HSS)
FTF	0.40
BPFO	3.58
BSF	4.67
BPFI	5.42

**Table 2 sensors-25-00815-t002:** RMS percentage of error in estimating RPM.

	Current RMSE	Vibration RMSE		
		Normal	Inner Race	Outer Race
Band A	4.64	58.82	3.59	3.84
Band B	1.01	1.42	1.41	0.92
Band C	1.05	2.13	1.26	0.98

## Data Availability

The data used in this work is a public dataset that can be accessed using the following link: https://data.mendeley.com/datasets/ztmf3m7h5x/6 (accessed on 1 October 2024).

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
