# Peer review of "Improved Order Tracking in Vibration Data Utilizing Variable Frequency Drive Signature"

_sensors, 2025, doi:10.3390/s25030815_

Round 1

Reviewer 1 Report

Comments and Suggestions for Authors

Reviewer comments:

The paper presents a signal processing technique aimed at improving order tracking in vibration data by utilizing Variable Frequency Drive (VFD) signatures. The algorithm was tested on a dataset to demonstrate the feasibility of using VFD signatures for accurate speed profile extraction and effective order tracking, leading to the identification of bearing faults. However, there are some points should be considered as follows:

1.    The manuscript's schematic representation of the speed measurement principle appears overly simplistic. It is recommended that the authors incorporate detailed illustrations or pictures.

2.    On the page 17, the authors stated that “Notably, Bands B and C exhibit symmetry with similar energy levels, in contrast to the scenario with the inner race defect.” What accounts for the disparity in spectral symmetry observed between inner and outer ring bearing faults?

3.    There should be a mistake in Figure 8d on line 238 of page 8

4. How does the proposed speed measurement method compare to existing technologies in terms of superiority or inferiority? Please provide a concise analysis and highlight the innovations of this study.

Author Response

The author would like to express sincere thanks to the reviewer for dedicating time and effort to review this manuscript. The valuable feedback provided has greatly contributed to improving the quality of the work. Below is a detailed response to the comments (also can be seen in the attached file)

  1. Comment: The manuscript's schematic representation of the speed measurement principle appears overly simplistic. It is recommended that the authors incorporate detailed illustrations or pictures.

Response:

Thank you for this suggestion. The publicly available dataset used in this study does not provide detailed information on the setup used for data collection, only indicating that a tachometer (Autonics FD-620-10) was used for speed measurement. In response, I have added more detail about the speed data collection in the manuscript to provide better clarity.

The revised version now includes the following addition (Line 243-245: revised manuscript): "The speed data was collected using a tachometer (Autonics FD-620-10) positioned on the HSS shaft (though this is not clearly shown in Figure 2). Instead of raw once-per-revolution data, RPM values for the HSS were provided at 0.11-second intervals."

For speed extraction algorithm illustrated in Figure 1, the following extra explanation has been added (lines 76-94 of the revised manuscript) The “pseudo-tacho” algorithm relies on discrete Fast Fourier Transform (FFT)and inverse FFT operations, making it practical and easily adaptable for industrial applications. The DFT transforms a time-domain signal x[n] into its frequency-domain representation X[k] using equation 1:

   , k=0,1,… ,N-1   (1)

Where:
-X[k]: Frequency-domain representation of the signal
-[n]: Time-domain signal
N: Total number of samples (FFT size)
k: Frequency index (0 to N-1)
n: Time sample index (0 to N-1)
j: Imaginary unit (sqrt(-1))

The inverse FFT (IFFT) reconstructs the time-domain signal from its frequency components as provided by equation 2:

             n=0,1,… ,N-1   (2)

These computations are efficiently performed using software tools like MATLAB, which provide built-in functions for FFT and IFFT, ensuring accurate and fast implementation.

In addition to this, Figure 9 (schematic presentation of the algorithm) has been updated to clearly differentiate the methodologies for speed estimation based on the two types of data. I explained how rotational speed is extracted from vibration signals using 2×VFD sidebands and compare this with the method used when current data is available. This distinction will help readers understand the adaptation of the algorithm depending on the data source and highlight the advantages of each approach in different scenarios.

The description of Figure 9 (lines 333-336) now reads “This process is illustrated schematically in Figure 9 in the cases of using either current or vibration data. Note that in the case of using vibration data, the selected band represents a sideband at 2×VFD and thus 2×VFD is directly calculated by dividing the sampling frequency by the sample difference between the zero crossings.”

  1. Comment: On page 17, the authors state, “Notably, Bands B and C exhibit symmetry with similar energy levels, in contrast to the scenario with the inner race defect.” What accounts for the disparity in spectral symmetry observed between inner and outer ring bearing faults?

Response:

Thank you for this comment. In response, I have expanded the explanation regarding the spectral symmetry observed in the presence of inner and outer race faults. This issue was briefly mentioned earlier in the manuscript (page 15, lines 413-419 :revised manuscript), but I have now provided a more detailed explanation to clarify the underlying causes of the disparity in spectral symmetry.

The following additional explanation has been incorporated into the manuscript on page 21, lines 500-510: "The disparity in spectral symmetry observed between the inner and outer race bearing faults may be explained by the different ways the vibration signals are influenced in each case. When there is spalling in the inner or outer race, impacts from the rolling elements at the spall location may excite different natural frequencies, which could alter the vibration spectrum. Specifically, for the inner race defect, the spall may move in and out of the load zone as the bearing rotates, causing modulations in the vibration signal. This interaction may generate more pronounced and uneven sidebands around the VFD carrier frequency. In contrast, for the outer race defect, the spall is typically stationary and often remains in the load zone, leading to a more continuous interaction with the rolling elements. This may result in more symmetrical sidebands and more stable energy levels around the VFD carrier frequency."

  1. Comment: There seems to be a mistake in Figure 8d on line 238 of page 8.

Response:

Thank you for pointing out the mismatch in the figure reference. I have corrected this mistake, and the manuscript now correctly refers to Figure 8d (line 361 of page 12).

  1. Comment: How does the proposed speed measurement method compare to existing technologies in terms of superiority or inferiority? Please provide a concise analysis and highlight the innovations of this study.

Response:

Thank you for your insightful comment. The primary contribution of this paper is the demonstration of the feasibility of using VFD sidebands for order tracking and bearing fault diagnosis, with a focus on how VFD sidebands can offer a non-intrusive and cost-effective solution. While this paper does not provide an exhaustive comparison with other technologies, I have included more context on the advantages of the proposed method over traditional techniques.

The following clarification has been added to the revised manuscript (line 210-226): "The primary contribution of this paper is demonstrating the feasibility of using VFD sidebands for order tracking and fault diagnosis, rather than providing an exhaustive comparison to other technologies. Benchmarking was performed against the actual rpm profile, which ensures a high level of confidence in the results. The paper focuses primarily on testing the use of a single frequency band around the VFD carrier to:

  1. Extract the rotor rpm profile (under light or no-load conditions).
  2. Utilize the extracted pseudo-tachometer information for order tracking (to remove speed fluctuations) and generate an order-tracked squared envelope spectrum for diagnosing bearing faults.
  3. In the first part of the analysis, the extracted, smoothened rpm profile using VFD sidebands (A and B) was compared to the actual rpm profile and spectrograms. The extracted rpm profile closely matched the actual rotor rpm profile, both in amplitude and shape. This demonstrates that VFD sidebands in the vibration signal can effectively extract an accurate rotor speed profile. Additionally, the order-tracked squared envelope spectrum, based on the VFD-extracted speed reference signals, clearly identified ball pass frequencies for both inner and outer race faults, which closely aligned with the calculated defect frequencies."

I have also updated section 4 (Discussion, Conclusions and Future Work) by adding the following (lines 567-578 of the revised manuscript)

“The method was validated by comparing the extracted speed profile using VFD sidebands against the actual RPM profile, ensuring a high level of confidence in the results. The extracted smoothened RPM profile closely matched the actual rotor RPM profile, both in amplitude and shape, which demonstrates that VFD sidebands can provide accurate rotor speed extraction. Additionally, the order-tracked squared envelope spectrum based on VFD-extracted speed reference signals clearly identified ball pass frequencies for both inner and outer race defects, closely matching the calculated defect frequencies.

The VFD carrier sidebands method is particularly effective because it offers a low-cost, non-intrusive solution that performs well under light load conditions, without the need for additional sensors. This makes it easier and more cost-effective to implement compared to more complex methods, which is why a direct comparison with other technologies was not the primary focus of this study.”

Thank you once again for your thoughtful and constructive feedback. Your comments have been extremely helpful in improving the clarity and quality o

Reviewer 2 Report

Comments and Suggestions for Authors

With interest, I've got acquainted with the manuscript dedicated to tacho-less estimation of the induction motor shaft rotation speed, which is required for subsequent diagnosis e.g. for bearing faults analysis. The paper is well and clearly written, qualitatively illustrated, and the plots are finely designed. The reference and description of the dataset work for overall good impression. The conclusions are generally supported by the results.

However, a number of questions and remarks arise while reading the manuscript.

 1. In Introduction, it should be mentioned that, in addition to "tacho-less" order tracking methods, there are methods for diagnosing and analyzing bearing faults which do not require speed data, which is essential especially at higher slip levels.  These include entropy-based methods, as well as recent bearing-fault detection method using return map analysis, a method based on approximation of wavelet coefficients by a generalized Gaussian distribution and others. Also, in Introduction, the contribution of the work should be strengthened by emphasizing the difference between the current work and the method described in [8].

2. The paper lacks mathematical apparatus to clarify the details of the algorithm. I recommend writing a step-by-step description, with the formulae of all transforms used, and the averaging formula. In particular, the expression for transforming the signal from frequency to order domain using the available velocity data. Also, formulae for bearing fault detection are appreciated.

3. I recommend that a brief description be given of the method of diagnosing bearing faults by envelopes, despite it is widely spread. In this brief, it is important to point the limitations of this method that encourage the development of alternative algorithms.

4. The algorithm diagram should emphasize the difference in calculating the rotational speed depending on whether current or vibration data is used.

5. The work does not compare the proposed method with others described in the literature (referenced in the Introduction) on the dataset under study, whether these methods are based on spectrograms, discrete spectrum correction or probability density functions. Hence, it is not clear whether the proposed method is more efficient compared to them.

6. From the results, it is not clear which frequency band, B or C, should be favored and in what conditions. Alternatively, is it worth to use a composite method using 2 or 3 bands?

With that, I believe that this promising manuscript may be accepted after revision.

Comments on the Quality of English Language

I recommend writing 'RPM' instead of 'rpm'.

Author Response

I greatly appreciate your thoughtful and constructive feedback on this manuscript, which focuses on tacho-less estimation of the induction motor shaft rotation speed for subsequent bearing fault diagnosis. I'm glad to know that the paper is clear, well-written, and visually supported by well-designed plots. It is rewarding to know that the reference and description of the dataset worked well and that the conclusions are generally supported by the results. Your feedback is invaluable, and I have provided detailed responses to each of your comments below (Also can be seen in the attached file)

  1. In Introduction, it should be mentioned that, in addition to "tacho-less" order tracking methods, there are methods for diagnosing and analysing bearing faults which do not require speed data, which is essential especially at higher slip levels.  These include entropy-based methods, as well as recent bearing-fault detection method using return map analysis, a method based on approximation of wavelet coefficients by a generalized Gaussian distribution and others. Also, in Introduction, the contribution of the work should be strengthened by emphasizing the difference between the current work and the method described in [8].

Response:

Thank you for your insightful comment. I agree that it is important to mention alternative methods for diagnosing and analyzing bearing faults such as entropy-based methods, return map analysis, and wavelet coefficient approximation methods. In response, I have added references to entropy-based methods and return map analysis. This expands the context of the current work and provides readers with a broader view of the available diagnostic techniques.

The following sentences have been added between (lines 112-119 of the revised manuscript). References 21-24 have been also added to the refence list.

In addition to "tacho-less" order tracking methods, various techniques exist for diagnosing and analyzing bearing faults without speed data, which is particularly important at higher slip levels. Entropy-based approaches, such as those presented by Leite et al. [21], offer effective solutions for early fault detection. Recent advancements in bearing fault detection also include return map analysis, which utilizes time-series feature extraction enhanced with wavelet transform techniques, as demonstrated by Ponomareva et al. [22]. These methods enable accurate detection and analysis of bearing faults even in the absence of direct speed data.

[21] Leite, G.D.N.P.; Araújo, A.M.; Rosas, P.A.C.; Stosic, T.; Stosic, B. Entropy Measures for Early Detection of Bearing Faults. Physica A: Statistical Mechanics and its Applications 2019, 514, 458–472.

[22] Ponomareva, V.; Druzhina, O.; Logunov, O.; Rudnitskaya, A.; Bobrova, Y.; Andreev, V.; Karimov, T. Time-Series Feature Extraction by Return Map Analysis and Its Application to Bearing-Fault Detection. Big Data Cogn. Comput. 2024, 8, 82.

Regarding the comparison with the method described in [8], I have strengthened the manuscript by clearly highlighting the contribution of the proposed approach. Specifically, I emphasized the use of VFD sidebands for speed estimation and order tracking, which distinguishes our method from others, especially when addressing tachometer-less challenges under variable load conditions. These revisions clarify the contribution of the current work and position it effectively within the broader context of bearing fault diagnostics. The following clarification has been added to the revised manuscript (line 210-226): "The primary contribution of this paper is demonstrating the feasibility of using VFD sidebands for order tracking and fault diagnosis, rather than providing an exhaustive comparison to other technologies. Benchmarking was performed against the actual rpm profile, which ensures a high level of confidence in the results. The paper focuses primarily on testing the use of a single frequency band around the VFD carrier to:

  1. Extract the rotor rpm profile (under light or no-load conditions).
  2. Utilize the extracted pseudo-tachometer information for order tracking (to remove speed fluctuations) and generate an order-tracked squared envelope spectrum for diagnosing bearing faults.
  3. In the first part of the analysis, the extracted, smoothened rpm profile using VFD sidebands (A and B) was compared to the actual rpm profile and spectrograms. The extracted rpm profile closely matched the actual rotor rpm profile, both in amplitude and shape. This demonstrates that VFD sidebands in the vibration signal can effectively extract an accurate rotor speed profile. Additionally, the order-tracked squared envelope spectrum, based on the VFD-extracted speed reference signals, clearly identified ball pass frequencies for both inner and outer race faults, which closely aligned with the calculated defect frequencies."

  1. The paper lacks mathematical apparatus to clarify the details of the algorithm. I recommend writing a step-by-step description, with the formulae of all transforms used, and the averaging formula. In particular, the expression for transforming the signal from frequency to order domain using the available velocity data. Also, formulae for bearing fault detection are appreciated.

Response:

Thank you for this valuable suggestion. To enhance the clarity of the algorithm, I have revised the manuscript to include extra description of the algorithm, along with the relevant mathematical equations.

The revised version now includes the following addition (Line 243-245: revised manuscript): "The speed data was collected using a tachometer (Autonics FD-620-10) positioned on the HSS shaft (though this is not clearly shown in Figure 2). Instead of raw once-per-revolution data, RPM values for the HSS were provided at 0.11-second intervals."

For speed extraction algorithm illustrated in Figure 1, the following extra explanations has bee added (lines 76-94 of the revised manuscript) The “pseudo-tacho” algorithm relies on discrete Fast Fourier Transform (FFT)and inverse FFT operations, making it practical and easily adaptable for industrial applications. The DFT transforms a time-domain signal x[n] into its frequency-domain representation X[k] using equation 1:

   , k=0,1,… ,N-1   (1)

Where:
-X[k]: Frequency-domain representation of the signal
-[n]: Time-domain signal
N: Total number of samples (FFT size)
k: Frequency index (0 to N-1)
n: Time sample index (0 to N-1)
j: Imaginary unit (sqrt(-1))

The inverse FFT (IFFT) reconstructs the time-domain signal from its frequency components as provided by equation 2:

             n=0,1,… ,N-1   (2)

In addition, order tracking purpose, has been better explained when using vibration signal processing. The following paragraph (lines 32-44) has been updated to replace previous first paragraph in the introduction:

“Frequency analysis of vibration signals is commonly used to evaluate the condition of rotating machinery. For this analysis to yield meaningful results, each vibration component must correspond to a distinct frequency. In rotating machinery, most vibration components are dependent on the speed of the drive shaft. When the shaft speed is constant, discrete frequency values are observed. However, when shaft speed fluctuates, frequency analysis becomes ineffective, as discrete frequency values no longer exist. To overcome this, the vibration signal can be re-sampled from the time domain to the phase domain. Since the shaft rotates the same angular distance in each revolution, re-sampling the signal to phase ensures that vibration components always correspond to discrete frequency values, regardless of speed variations. This transformation produces an order-spectrum, where frequency analysis remains valid even for machines with variable speeds. The technique used to generate this order-spectrum is known as order-tracking [1].”

Equations to calculate bearing defect frequencies were added (lines 280-298 of the revised manuscript) as follows:

Equations (4-7) were used to calculate bearing defect frequencies for various components as follows:

Ball Pass Frequency Inner Race (BPFI): Frequency at which balls pass a point on the inner race.

                              (4)

Ball Pass Frequency Outer Race (BPFO): Frequency at which balls pass a point on the outer race.

                             (5)

Ball Spin Frequency (BSF): Rotational frequency of the rolling elements themselves.

          (6)

Fundamental Train Frequency (FTF): Frequency at which the cage rotates relative to the bearing.

                            (7)

Where:
 n: Number of rolling elements
 d: Diameter of rolling elements
 D: Pitch diameter of the bearing
 : Contact angle (in radians)

: shaft rotational speed (Hz, rpm or 1 for orders)

  1. I recommend that a brief description be given of the method of diagnosing bearing faults by envelopes, despite it is widely spread. In this brief, it is important to point the limitations of this method that encourage the development of alternative algorithms.

Response

I appreciate your suggestion to include a brief description of the envelope analysis method. While widely used, I agree that it is important to provide a background to the reader.  I have added a concise overview of the envelope analysis method in the introduction (New section: section 1.3: lines 158-199), pointing out its common use in bearing fault diagnosis. Two references have been used and an extra summary Figure (Figure 2 in the revised manuscript has been added)

  1. The algorithm diagram should emphasize the difference in calculating the rotational speed depending on whether current or vibration data is used.

Response

Thank you for this valuable comment. I understand the need to emphasize the difference in calculating rotational speed depending on whether current or vibration data is used. In response,

 I have now updated the algorithm diagram (now Figure 9 of the updated manuscript) to clearly differentiate the methodologies for speed estimation based on the two types of data. I explained how rotational speed is extracted from vibration signals using 2×VFD sidebands and compare this with the method used when current data is available. This distinction will help readers understand the adaptation of the algorithm depending on the data source and highlight the advantages of each approach in different scenarios.

The description of Figure 9 (lines 373-376) now reads “This process is illustrated schematically in Figure 9 in the cases of using either current or vibration data. Note that in the case of using vibration data, the selected band represents a sideband at 2×VFD and thus 2×VFD is directly calculated by dividing the sampling frequency by the sample difference between the zero crossings.”

  1. The work does not compare the proposed method with others described in the literature (referenced in the Introduction) on the dataset under study, whether these methods are based on spectrograms, discrete spectrum correction or probability density functions. Hence, it is not clear whether the proposed method is more efficient compared to them.

Response

Thank you for pointing this out. The main contribution of this paper is demonstrating the feasibility of using VFD sidebands for order tracking and bearing fault diagnosis, which is why a detailed comparison with other methods is not the primary focus. However, I have now added a section that clarifies the contribution of the paper (section 1.4) and updated section 4 (Discussion, Conclusions and Future Work) by adding the following (lines 567-578 of the revised manuscript)

“The method was validated by comparing the extracted speed profile using VFD sidebands against the actual RPM profile, ensuring a high level of confidence in the results. The extracted smoothened RPM profile closely matched the actual rotor RPM profile, both in amplitude and shape, which demonstrates that VFD sidebands can provide accurate rotor speed extraction. Additionally, the order-tracked squared envelope spectrum based on VFD-extracted speed reference signals clearly identified ball pass frequencies for both inner and outer race defects, closely matching the calculated defect frequencies.

The VFD carrier sidebands method is particularly effective because it offers a low-cost, non-intrusive solution that performs well under light load conditions, without the need for additional sensors. This makes it easier and more cost-effective to implement compared to more complex methods, which is why a direct comparison with other technologies was not the primary focus of this study.”

  1. From the results, it is not clear which frequency band, B or C, should be favored and in what conditions. Alternatively, is it worth to use a composite method using 2 or 3 bands?

Response

Thank you for raising this important point. In response, this issue has been addressed earlier in the manuscript (page 15, lines 413-419: revised manuscript). The following additional explanation has been added to page 21, lines 500-510:

The disparity in spectral symmetry observed between inner and outer race bearing faults can be explained by the different ways the vibration signals are influenced in each case. When spalling occurs in the inner or outer race, impacts from the rolling elements at the spall location may excite different natural frequencies, which could alter the vibration spectrum. Specifically, for the inner race defect, the spall may move in and out of the load zone as the bearing rotates, causing modulations in the vibration signal. This interaction can generate more pronounced and uneven sidebands around the VFD carrier frequency. In contrast, for the outer race defect, the spall is typically stationary and often remains within the load zone, leading to a more continuous interaction with the rolling elements. This may result in more symmetrical sidebands and similar energy levels around the VFD carrier frequency. The use of either Bands B or C around the VFD carrier may then vary on a case-by-case basis, depending on the dynamic characteristics of the system. As a result, no general rule can be given for when to use Band B or Band C. Additionally, a composite method utilizing both bands is not feasible, as the algorithm relies primarily on the use of a single frequency band to extract the speed reference.

Thank you once again for your insightful comments. The revisions made in response to your suggestions should enhance the clarity and robustness of the manuscript. I believe the revised version will be significantly strengthened and will address all of your concerns.

Round 2

Reviewer 2 Report

Comments and Suggestions for Authors

Dear Author! I read your detailed and deep responses to my comments. The fine work was done, which undoubtedly improved the quality of the manuscript.

However, unfortunately, you attached the old version of the manuscript as version 2, so I was not able to get acquainted with the updated version of the paper. Please attach the correct version.

Author Response

Dear Reviewer,
Thank you very much for reviewing my manuscript and for your comments.
I have now attached the updated manuscript for your consideration.

Kind regards,

Nader

Round 3

Reviewer 2 Report

Comments and Suggestions for Authors

With pleasure, I’ve got acquainted with the renewed version of the manuscript. The work done is solid, and the manuscript have got significantly improved. Minor remarks include:

1. In Section 1.3, not only benefits, but also limitations of the envelope analysis method should be presented.

2. I recommend inserting plots of the speed measurement error in each band relative to the exact values measured by the tachometer. The resulting estimate  (in %) after being averaged should be compared to the accuracy of other order tracking methods from the referenced literature. It will also be useful to readers for the same purpose.

3. Also, I recommend separating Conclusion section, which should present a short overview of the work done, with no deep details. In Conclusion, the RMS error of the calculated rotation speed (in %) obtained in the most appropriate band should be given, as well as other important numerical estimates, and contributions and limitations. 

4. Some typos were found.

a) ‘rpm’ and other abbreviations should be spelled using caps letters, i.e. ‘RPM’

b) In Line 226, symbol “ is redundant.

Author Response

Author's Reply to the Review Report (Reviewer 2)

I sincerely thank the reviewer for their thoughtful comments and valuable feedback, which have significantly enhanced the paper's readability and clarity. Below is a detailed response to the latest comments. In the revised manuscript, updates for this round are highlighted in blue, while updates from the previous round remain in red to illustrate the paper's progression.

  1. In Section 1.3, not only benefits, but also limitations of the envelope analysis method should be presented.

Response:

Comment acknowledged: Thank you for your feedback. I have revised the manuscript by adding a paragraph (last paragraph: lines 194–200) in Section 1.4.

Envelope analysis works well in cases of incipient faults and rolling contact fatigue (RCF), which produce impulses in the system. It requires the identification of a suitable band for demodulation. However, the absence of RCF limits its effectiveness, as bearing defect frequencies are not expected to appear in the envelope spectrum. In such cases, statistical health indicators relying on signal impulsiveness, such as Kurtosis, are not suitable. Instead, monitoring friction level increases, noise, and energy changes in the system would be more appropriate.

  1. I recommend inserting plots of the speed measurement error in each band relative to the exact values measured by the tachometer. The resulting estimate (in %) after being averaged should be compared to the accuracy of other order tracking methods from the referenced literature. It will also be useful to readers for the same purpose.

Response:

Thank you for this valuable suggestion.

I have now calculated the root Mean Square error (RMSE) and inserted four figures (Figure 10, 16,22 and 27). The text describing the calculation of the relative error and comments on each of these figures have been added in lines (370-385), (439-446) , (502-507) and (562-565).

In addition, a summary table (Table 2) of all  RMSE values were created and added to the conclusion section with proper comments.

  1. Also, I recommend separating Conclusion section, which should present a short overview of the work done, with no deep details. In Conclusion, the RMS error of the calculated rotation speed (in %) obtained in the most appropriate band should be given, as well as other important numerical estimates, and contributions and limitations.

Response

I appreciate your suggestion to separate the conclusion section and provide a summary of the % errors.

I have restructured the paper to have a summary and discussion section followed by Conclusions and Future Work. The conclusion now contains a summary table of the RMSE which compares different cases and the use of different bands. Results are discussed and future work is provided to further investigate the potential of improving the proposed algorithm and test it under load conditions.

I have also updated the abstract to show the RMSE results obtained from using the bands around the VFD carrier to extract a tachometer/speed reference.

  1. Some typos were found.
  2. a) ‘rpm’ and other abbreviations should be spelled using caps letters, i.e. ‘RPM’
  3. b) In Line 226, symbol “ is redundant.

Response

Comment Acknowledged.

  • rpm changed into RPM in both text and Figures.
  • Redundant symbol “removed
